# Deterministic Symmetric Positive Semidefinite Matrix Completion

**William E. Bishop**[1,2]**, Byron M. Yu**[2,3,4]
[1]Machine Learning, [2]Center for the Neural Basis of Cognition,
[3]Biomedical Engineering, [4]Electrical and Computer Engineering
Carnegie Mellon University
{wbishop, byronyu}@cmu.edu

## Abstract

We consider the problem of recovering a symmetric, positive semidefinite (SPSD) matrix from a subset of its entries, possibly corrupted by noise. In contrast to previous matrix recovery work, we drop the assumption of a random sampling of entries in favor of a deterministic sampling of principal submatrices of the matrix. We develop a set of sufficient conditions for the recovery of a SPSD matrix from a set of its principal submatrices, present necessity results based on this set of conditions and develop an algorithm that can exactly recover a matrix when these conditions are met. The proposed algorithm is naturally generalized to the problem of noisy matrix recovery, and we provide a worst-case bound on reconstruction error for this scenario. Finally, we demonstrate the algorithm's utility on noiseless and noisy simulated datasets.

## 1 Introduction

There are multiple scenarios where we might wish to reconstruct a symmetric positive semidefinite (SPSD) matrix from a sampling of its entries. In multidimensional scaling, for example, pairwise distance measurements are used to form a kernel matrix and PCA is performed on this matrix to embed the data in a low-dimensional subspace. However, due to constraints, it may not be possible to measure pairwise distances for all variables, rendering the kernel matrix incomplete. In neuroscience a population of neurons is often modeled as driven by a low-dimensional latent state [1], producing a low-rank covariance structure in the observed neural recordings. However, with current technology, it may only be possible to record from a large population of neurons in small, overlapping sets [2,3], leaving holes in the empirical covariance matrix. More generally, SPSD matrices in the form of Gram matrices play a key role in a broad range of machine learning problems such as support vector machines [4], Gaussian processes [5] and nonlinear dimensionality reduction techniques [6] and the reconstruction of such matrices from a subset of their entries is of general interest.

In real world scenarios, the constraints that make it difficult to observe a whole matrix often also constrain which particular entries of a matrix are observable. In such settings, existing matrix completion results, which assume matrix entries are revealed in an unstructured, random manner [7–14] or the ability to finely query individual entries of a matrix in an adaptive manner [15, 16] might not be applicable. This motivates us to examine the problem of recovering a SPSD matrix from a given, deterministic set of its entries. In particular we focus on reconstructing a SPSD matrix from a revealed set of its principal submatrices.

Recall that a principal submatrix of a matrix is a submatrix obtained by symmetrically removing rows and columns of the original matrix. When individual entries of a matrix are formed by pairwise measurements between experimental variables, principal submatrices are a natural way to formally capture how entries are revealed.

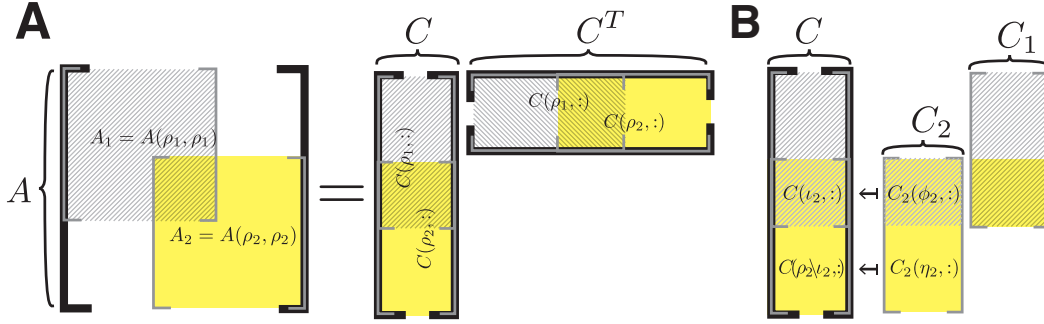

Figure 1: (A) An example $A$ matrix with two principal submatrices, showing the correspondence between $A(\rho_l, \rho_l)$ and $C(\rho_l, :)$. (B) Mapping of $C_1$ and $C_2$ to $C$, illustrating the role of $\iota_l$, $\phi_l$ and $\eta_l$.

Sampling principal submatrices also allows for an intuitive method of matrix reconstruction. As shown in Fig. 1, any $n \times n$ rank $r$ SPSD matrix $A$ can be decomposed as $A = CC^T$ for some $C \in \mathbb{R}^{n \times r}$. Any principal submatrix of $A$ can also be decomposed in the same way. Further, if $\rho_i$ is an ordered set indexing the the $i^{th}$ principal submatrix of $A$, it must be that $A(\rho_i, \rho_i) = C(\rho_i, :)C(\rho_i, :)^T$.[1] This suggests we can decompose each $A(\rho_i, \rho_i)$ to learn the the rows of $C$ and then reconstruct $A$ from the learned $C$, but there is one complication. Any matrix, $C(\rho_i, :)$, such that $A(\rho_i, \rho_i) = C(\rho_i, :)C(\rho_i, :)^T$, is only defined up to an orthonormal transformation. The naïve algorithm just suggested has no way of ensuring the rows of $C$ learned from two different principal submatrices are consistent with respect to this degeneracy. Fortunately, the situation is easily remedied if the principal submatrices in question have some overlap, so that the $C(\rho_i, :)$ matrices have some rows that map to each other. Under appropriate conditions explored below, we can learn unique orthonormal transformations rendering these rows equal, allowing us to align the $C(\rho_i, :)$ matrices to learn a proper $C$.

**Contributions**   In this paper, we make the following contributions.

1. We prove sufficient conditions, which are also necessary in certain situations, for the exact recovery of a SPSD matrix from a given set of its principal submatrices.

2. We present a novel algorithm which exactly recovers a SPSD matrix when the sufficient conditions are met.

3. The algorithm is generalized when the set of observed principal submatrices of a matrix are corrupted by noise. We present a theorem guaranteeing a bound on reconstruction error.

## 1.1   Related Work

The low rank matrix completion problem has received considerable attention since the work of Candès and Recht [17] who demonstrated that a simple convex problem could exactly recover many low-rank matrices with high probability. This work, as did much of what followed (e.g., [7–9]), made three key assumptions. First, entries of a matrix were assumed to be uncorrupted by noise and, second, revealed in a random, unstructured manner. Finally, requirements, such as incoherence, were also imposed to rule out matrices with most of their mass concentrated in a only a few entries.

These assumptions have been reexamined and relaxed in additional work. The case of noisy observed entries has been considered in [10–14]. Others have reduced or removed the requirements for incoherence by using iterative, adaptive sampling schemes [15, 16]. Finally, recent work [18, 19] has considered the case of matrix recovery when entries are selected a deterministic manner.

Our work considerably differs from this earlier work. Our applications of interest allow us to assume much structure, i.e., that matrices are SPSD, which our algorithm exploits, and our sufficient conditions make no appeal to incoherence. Our work also differs from previous results for deterministic sampling schemes (e.g., [18, 19]), which do not consider noise nor provide sufficient conditions for exact recovery, instead approaching the problem as one of matrix approximation.

Previous work has also considered the problem of completing SPSD matrices of any [20] or low rank [21,22]. Our work to identify conditions for a unique completion of a given rank can be viewed as a continuation of this work where our sufficient and necessary conditions can be understood in a particularly intuitive manner due to our sampling scheme. Finally, the Nyström method [23] is a well known technique for approximating a SPSD matrix as low rank. It can also be applied to the matrix recovery problem, and in the noiseless case, sufficient conditions for exact recovery are known [24]. However, the Nyström method requires sampling full columns and rows of the original matrix, a sampling scheme which may not be possible in many of our applications of interest.

## 2   Preliminaries

### 2.1   Deterministic Sampling for SPSD Matrices

We denote the set of index pairs for the revealed entries of a matrix by $\Omega$. Formally, an index pair, $(i, j)$, is in $\Omega$ if and only if we observe the corresponding entry of an $n \times n$ matrix so that $\Omega \subset [n] \times [n]$.[2] In this work, we assume $\Omega$ indexes a set of principal submatrices of a matrix. Let $\Omega_l \subseteq \Omega$ indicate a subset of $\Omega$. If $\Omega_l$ indexes a principal submatrix of a matrix, it can be compactly described by the unique set of row (or equivalently column) indices it contains. Let $\boldsymbol{\rho}\{\Omega_l\} = \{i | (i, j) \in \Omega_l\}$ be the set of row indices contained in $\Omega_l$. For compactness, let $\rho_l = \boldsymbol{\rho}\{\Omega_l\}$. Finally, let $|\cdot|$ indicate cardinality. Then, for an $n \times n$ matrix, $A$, of rank $r$ we make the following assumptions on $\Omega$.

> **(A1)** $\boldsymbol{\rho}\{\Omega\} = [n]$.
>
> **(A2)** There exists a collection $\Omega_1, \ldots, \Omega_k$ of subsets of $\Omega$ such that $\Omega = \cup_{l=1}^k \Omega_l$, and for each $\Omega_l$, $(i, i) \in \Omega_l$ and $(j, j) \in \Omega_l$ if and only if $(i, j) \in \Omega_l$ and $(j, i) \in \Omega_l$.
>
> **(A3)** There exists a collection $\Omega_1, \ldots, \Omega_k$ of subsets of $\Omega$ such that $A2$ holds and if $k > 1$, there exists an ordering $\tau_1, \ldots, \tau_k$ such that for all $i \geq 2$, $|\rho_{\tau_i} \cap \left( \cup_{j=1}^{i-1} \rho_{\tau_j} \right)| \geq r$.

The first assumption ensures $\Omega$ indexes at least one entry for each row of $A$. Assumption $A2$ requires that $\Omega$ indexes a collection of principal submatrices of $A$, and $A3$ allows for the possible alignment of rows of $C$ (recall, $A = CC^T$) estimated from each principal submatrix.

### 2.2   Additional Notation

Denote the set of real, $n \times n$ SPSD matrices by $\mathcal{S}_+^n$, and let $A \in \mathcal{S}_+^n$ be the rank $r$ matrix to be recovered. For the noisy case, $\tilde{A}$ will indicate a perturbed version of $A$. We will use the notation $A_l$ to indicate the principal submatrix of a matrix $A$ indexed by $\Omega_l$.

Denote the eigendecomposition of $A$ as $A = E\Lambda E^T$ for the diagonal matrix $\Lambda \in \mathbb{R}^{r \times r}$ containing the non-zero eigenvalues of $A$, $\lambda_1 \geq \ldots \geq \lambda_r$, along its diagonal and the matrix $E^{n \times r}$ containing the corresponding eigenvectors of $A$ in its columns. Let $n_l$ denote the size of $A_l$ and $r_l$ the rank. Because $A_l$ is a principal submatrix of $A$, it follows that $A_l \in \mathcal{S}_+^{n_l}$. Denote the eigendecomposition of each $A_l$ as $A_l = E_l \Lambda_l E_l^T$ for the matrices $\Lambda_l \in \mathbb{R}^{r_l \times r_l}$ and $E_l \in \mathbb{R}^{n_l, r_l}$. We add tildes to the appropriate symbols for the eigendecomposition of $\tilde{A}$ and its principal submatrices.

Finally, let $\iota_l = \rho_{\tau_l} \cap (\cup_{j=1,\ldots,l-1} \rho_{\tau_j})$ be the intersection of the indices for the $l^{th}$ principal submatrix with the indices of the all of the principal submatrices ordered before it. Let $C_l$ be a matrix such that $C_l C_l^T = A_l$. If $A_l$ is a principal submatrix of $A$ there will exist some $C_l$ such that $C(\rho_l, :) = C_l$. For such a $C_l$, let $\phi_l$ be an index set that assigns the rows of the matrix $C(\iota_l, :)$ to their location in $C_l$, so that $C(\iota_l, :) = C_l(\phi_l, :)$ and let $\eta_l$ assign the rows of $C(\rho_l \setminus \iota_l, :)$ to their

**Algorithm 1** SPSD Matrix Recovery $(r, \{\tilde{E}_l, \tilde{\Lambda}_l, \tau_l, \rho_l, \phi_l, \iota_l, \eta_l\}_{l=1}^k)$

Initialize $\hat{C}$ as a $n \times r$ matrix.

    1. $\hat{C}(\rho_{\tau_1}, :) \leftarrow \tilde{E}_{\tau_1}(:, 1:r) \tilde{\Lambda}_{\tau_1}^{1/2}(1:r, 1:r)$

    2. For $l \in \{2, \ldots, k\}$

        (a) $\hat{C}_l \leftarrow \tilde{E}_{\tau_l}(:, 1:r) \tilde{\Lambda}_{\tau_l}^{1/2}(1:r, 1:r)$

        (b) $\hat{W}_l \leftarrow \operatorname{argmin}_{WW^T=I} \|\hat{C}(\iota_l, :) - \hat{C}_l(\phi_l, :)W\|_F^2$

        (c) $\hat{C}(\rho_{\tau_l} \setminus \iota_l, :) \leftarrow \hat{C}_l(\eta_l, :)\hat{W}_l$

    3. Return $\hat{A} = \hat{C}\hat{C}^T$

---

location in $C_l$, so that $C(\rho_l \setminus \iota_l, :) = C_l(\eta_l, :)$. The role of $\rho_l$, $\iota_l$, $\eta_l$ and $\phi_l$ is illustrated for the case of two principal submatrices with $\tau_1 = 1$, $\tau_2 = 2$ in Figure 1.

## 3 The Algorithm

Before establishing a set of sufficient conditions for exact matrix completion, we present our algorithm. Except for minor notational differences, the algorithms for the noiseless and noisy matrix recovery scenarios are identical, and for brevity we present the algorithm for the noisy scenario.

Let $\Omega$ sample the observed entires of $\tilde{A}$ so that $A1$ through $A3$ hold. Assume each perturbed principal submatrix, $\tilde{A}_l$, indexed by $\Omega$ is SPSD and of rank $r$ or greater. These assumptions on each $\tilde{A}_l$ will be further explored in section 5. Decompose each $\tilde{A}_l$ as $\tilde{A}_l = \tilde{E}_l \tilde{\Lambda}_l \tilde{E}_l^T$, and form a rank $r$ matrix $\hat{C}_l$ as $\hat{C}_l = \tilde{E}_l(:, 1:r) \tilde{\Lambda}_l^{1/2}(1:r, 1:r)$.

The rows of the $\hat{C}_l$ matrices contain estimates for the rows of $C$ such that $A = CC^T$, though rows estimated from different principal submatrices may be expressed with respect to different orthonormal transformations. Without loss of generality, assume the principal submatrices are labeled so that $\tau_1 = 1, \ldots, \tau_k = k$. Our algorithm begins to construct $\hat{C}$ by estimating $\hat{C}(\rho_1, :) = \hat{C}_1$. In this step, we also implicitly choose to express $\hat{C}$ with respect to the basis for $\hat{C}_1$. We then iteratively add rows to $\hat{C}$, for each $\hat{C}_l$ adding the rows $\hat{C}_l(\eta_l, :)$ to $\hat{C}$. To estimate the orthornormal transformation to align the rows of $\hat{C}_l$ with the rows of $\hat{C}$ estimated in previous iterations, we solve the following optimization problem

$$\hat{W}_l = \operatorname*{argmin}_{WW^T=I} \left\|\hat{C}(\iota_l, :) - \hat{C}_l(\phi_l, :)W\right\|_F^2. \tag{1}$$

In words, equation 1 estimates $\hat{W}_l$ so that the rows of $\hat{C}_l$ which overlap with the previously estimated rows of $\hat{C}$ match as closely as possible. In the noiseless case, (1) is equivalent to $\hat{W}_l = W : \hat{C}(\iota_i, :) - \hat{C}_l(\phi_i, :)W = 0$. Equation 1 is known as the Procrustes problem and is non-convex, but its solution can be found in closed form and sufficient conditions for its unique solution are known [25].

After learning $\hat{W}_l$ for each $\hat{C}_l$, we build up the estimate for $\hat{C}$ by setting $\hat{C}(\rho_l \setminus \iota_l, :) = \hat{C}_l(\eta_l, :)\hat{W}_l$. This step adds the rows of $\hat{C}_l$ that do not overlap with those already added to $\hat{C}$ to the growing estimate of $\hat{C}$. If we process principal submatrices in the order specified by $A3$, this algorithm will generate a complete estimate for $\hat{C}$. The full matrix $\hat{A}$ can then be estimated as $\hat{A} = \hat{C}\hat{C}$. The pseudocode for this algorithm is given in Algorithm 1.

## 4 The Noiseless Case

We begin this section by stating one additional assumption on $A$.

**(A4)** There exists a collection $\Omega_1, \ldots, \Omega_k$ of subsets of $\Omega$ such that $A2$ holds and if $k > 1$, there exists an ordering $\tau_1, \ldots, \tau_k$ such that the rank of $A(\iota_l, \iota_l)$ is equal to $r$ for each $l \in \{2, \ldots, k\}$.

In Theorem 2 we show that $A1$ - $A4$ are sufficient to guarantee the exact recovery of $A$. Conditions $A1$ - $A4$ can also be necessary for the unique recovery of $A$ by any method, as we show next in Theorem 1. Theorem 1 may at first glance appear quite simple, but it is a restatement of Lemma 6 in the appendix, from which more general necessity results can be derived. Specifically, Corollary 7 in the appendix can be used to establish the above conditions are necessary to recover $A$ from a set of its principal submatrices which can be aligned in a overlapping sequence (e.g., submatrices running down the diagonal of $A$), which might be encountered when constructing a covariance matrix from sequentially sampled subgroups of variables. Corollary 8 establishes a similar result when there exists a set of principal submatrices which have no overlap among themselves but all overlap with one other submatrix not in the set, and Corollary 9 establishes that it is sufficient to find just one principal submatrix that obeys certain conditions with respect to the rest of the sampled entries of the matrix to certify the impossibility of matrix completion. This last corollary in fact applies even when the rest of the sampled entries do not fall into a union of principal submatrices of the matrix.

**Theorem 1.** *Let $\Omega \neq [n] \times [n]$ index $A$ so that $A2$ holds for some $\Omega_1 \subseteq \Omega$ and $\Omega_2 \subseteq \Omega$. Then $A1$, $A3$ and $A4$ must hold with respect to $\Omega_1$ and $\Omega_2$ for $A$ to be recoverable by any method.*

The proof can be found in the appendix. Here we briefly provide the intuition. Key to understanding the proof is recognizing that recovering $A$ from the set of entries indexed by $\Omega$ is equivalent to learning a matrix $C$ from the same set of entries such that $A = CC^T$. If $A1$ is not met, a complete row and the corresponding column of $A$ is not sampled, and there is nothing to constrain the estimate for the corresponding row of $C$. If $A3$ and $A4$ are not met, we can construct a $C$ such that all of the entries of the matrices $A$ and $CC^T$ indexed by $\Omega$ are identical yet $A \neq CC^T$.

We now show that our algorithm can recover $A$ as soon as the above conditions are met, establishing their sufficiency.

**Theorem 2.** *Algorithm 1 will exactly recover $A$ from a set of its principal submatrices indexed by $\Omega_1, \ldots, \Omega_k$ which meets conditions $A1$ through $A4$.*

The proof, which is provided in the appendix, shows that in the noiseless case, for each principal submatrix, $A_l$, of A, step 2a of Algorithm 1 will learn an exact $\hat{C}_l$ such that $A_l = \hat{C}_l \hat{C}_l^T$. Further, when assumptions $A3$ and $A4$ are met, step 2b will correctly learn the orthonormal transformation to align each $\hat{C}_l$ to the previously added rows of $\hat{C}$. Therefore, progressive iterations of step 2 correctly learn more and more rows of a unified $\hat{C}$. As the algorithm progresses, all of the rows of $\hat{C}$ are learned and the entirety of $A$ can be recovered in step 3 of the algorithm.

It is instructive to ask what we have gained or lost by constraining ourselves to sampling principal submatrices. In particular, we can ask how many individual entries must be observed before we can recover a matrix. A SPSD matrix has at least $nr$ degrees of freedom, and we would not expect any matrix recovery method to succeed before at least this many entries of the original matrix are revealed. The next theorem establishes that our sampling scheme is not necessarily wasteful with respect to this bound.

**Theorem 3.** *For any rank $r \geq 1$ matrix $A \in \mathcal{S}_+^n$ there exists a $\Omega$ such that $A1 - A3$ hold and*

$$|\Omega| \leq n(2r + 1).$$

Of course, this work is motivated by real-world scenarios where we are not at the liberty to finely select the principal submatrices we sample, and in practice we may often have to settle for a set of principal submatrices which sample more of the matrix. However, it is reassuring to know that our sampling scheme does not necessarily require a wasteful number of samples.

We note that assumptions $A1$ through $A4$ have an important benefit with respect to a requirement of incoherence. Incoherence is an assumption about the entire row and column space of a matrix and cannot be verified to hold with only the observed entries of a matrix. However, assumptions $A1$ through $A4$ can be verified to hold for a matrix of known rank using its observed entries. Thus, it is possible to verify that these assumptions hold for a given $\Omega$ and $A$ and provide a certificate guaranteeing exact recovery before matrix completion is attempted.

# 5 The Noisy Case

We analyze the behavior of Algorithm 1 in the presence of noise. For simplicity, we assume each observed, noise corrupted principal submatrix is SPSD so that the eigendecompositions in steps 1 and 2a of the algorithm are well defined. In the noiseless case, to guarantee the uniqueness of $\hat{A}$, $A4$ required each $A(\iota_l, \iota_l)$ to be of rank $r$. In the noisy case, we place a similar requirement on $\tilde{A}(\iota_l, \iota_l)$, where we recognize that the rank of each $\tilde{A}(\iota_l, \iota_l)$ may be larger than $r$ due to noise.

> **(A5)** There exists a collection $\Omega_1, \ldots, \Omega_k$ of subsets of $\Omega$ such that $A2$ holds and if $k > 1$, there exists an ordering $\tau_1, \ldots, \tau_k$ such that the rank of $\tilde{A}(\iota_l, \iota_l)$ is greater than or equal to $r$ for each $l \in \{2, \ldots, k\}$.

> **(A6)** There exists a collection $\Omega_1, \ldots, \Omega_k$ of subsets of $\Omega$ such that $A2$ holds and $\tilde{A}_l \in \mathcal{S}_+^{n_l}$ for each $l \in \{1, \ldots, k\}$.

In practice, any $\tilde{A}_l$ which is not SPSD can be decomposed into the sum of a symmetric and an antisymmetric matrix. The negative eigenvalues of the symmetric matrix can then be set to zero, rendering a SPSD matrix. As long as this resulting matrix meets the rank requirement in $A5$, it can be used in place of $\tilde{A}_l$. Our algorithm can then be used without modification to estimate $\hat{A}$.

**Theorem 4.** *Let $\Omega$ index an $n \times n$ matrix $\tilde{A}$ which is a perturbed version of the rank $r$ matrix $A$ such that $A1 - A6$ simultaneously hold for a collection of principal submatrices indexed by $\Omega_1, \ldots, \Omega_k$. Let $b \geq \max_{l \in [k]} \|C_l\|_F$ for some $C_l \in \mathbb{R}^{n_l \times r}$ such that $A_l = C_l C_l^T$, $\zeta \geq \lambda_{l,1}$, and $\delta \leq \min\{\min_{i \in [r-1]}, |\lambda_{l,i} - \lambda_{l,i+1}|, \lambda_{l,r}\}$. Assume $\|A_l - \tilde{A}_l\|_F \leq \epsilon$ for all $l$ for some $\epsilon < \min\{b^2/r, \delta/2, 1\}$. Then if in step 2 of Algorithm 1, $\mathbf{rank}\left\{\hat{C}_l(\phi_l, :)^T \hat{C}(\iota_l, :)\right\} = r$ for all $l \geq 2$, Algorithm 1 will estimate an $\hat{A}$ from the set of principal submatrices of $\tilde{A}$ indexed by $\Omega$ such that*

$$\left\|A - \hat{A}\right\|_F \leq 2G^{k-1}L\|C\|_F\sqrt{r}\epsilon + G^{2k-2}L^2 r\epsilon,$$

*where $C \in \mathbb{R}^{n \times r}$ is some matrix such that $A = CC^T$, $G = 4 + 12/v$, and $v \leq \lambda_r(A(\iota_r, \iota_r))/b^2$ for all $l$ and $L = \sqrt{1 + \frac{16\zeta}{\delta^2} + \frac{8\sqrt{2}\zeta^{1/2}}{\delta^{3/2}}}$.*

The proof is left to the appendix and is accomplished in two parts. In the first part, we guarantee that the ordered eigenvalues and eigenvectors of each $\tilde{A}_l$, which are the basis for estimating each $\hat{C}_l$, will not be too far from those of the corresponding $A_l$. In the second part, we bound the amount of additional error that can be introduced by learning imperfect $\hat{W}$ matrices which result in slight misalignments as each $\hat{C}_l$ matrix is incorporated into the final estimate for the complete $\hat{C}$. This second part relies on a general perturbation bound for the Procrustes problem, derived as Lemma 16 in the appendix.

Our error bound is non-probabilistic and applies in the presence of adversarial noise. While we know of no existing results for the recovery of matrices from deterministic samplings of noise corrupted entries, we can compare our work to bounds obtained for various results applicable to random sampling schemes, (e.g., [10–13]). These results require either incoherence [10, 11], boundedness [13] of the entries of the matrix to be recovered or assume the sampling scheme obeys the restricted isometry property [12]. Error is measured with various norms, but in all cases shows a linear dependence on the size of the original perturbation. For this initial analysis, our bound establishes that reconstruction error consistently goes to 0 with perturbation size, and we conjecture that with a refinement of our proof technique we can prove a linear dependence on $\epsilon$. We provide initial evidence for this conjecture in the results below.

# 6 Simulations

We demonstrate our algorithm's performance on simulated data, starting with the noiseless setting in Fig. 2. Fig. 2A shows three sampling schemes, referred to as masks, that meet assumptions $A1$

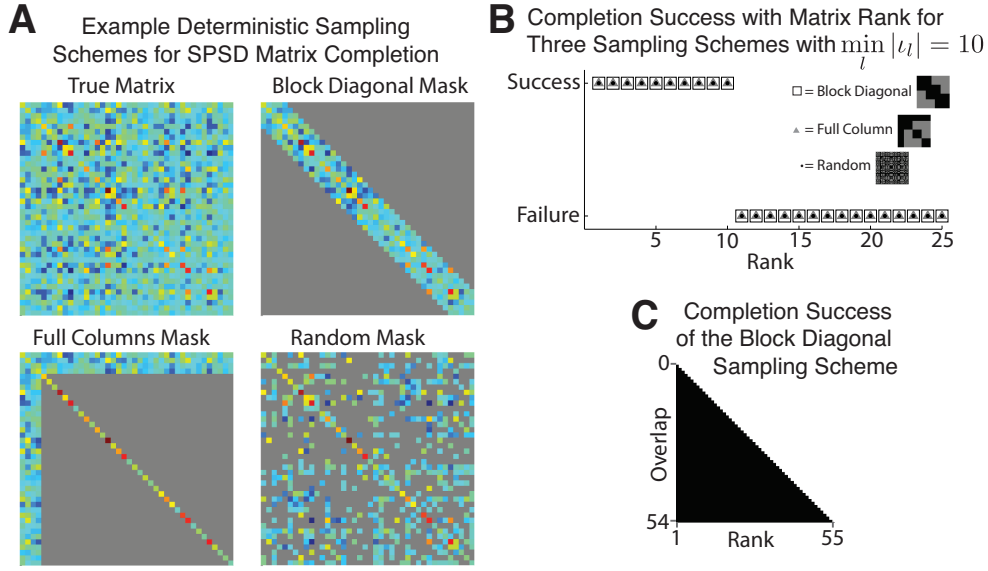

Figure 2: Noiseless simulation results. (A) Example masks for successful completion of a rank 4 matrix. (B) Completion success as rank is varied for masks with minimal overlap ($\min_l |\iota_l|$) of 10. (C) Completion success for rank $1 - 55$ matrices with block diagonal masks with minimal overlap ranging between $0 - 54$.

through $A3$ for a randomly generated $40 \times 40$ rank 4 matrix. In all of the noiseless simulations, we simulate a rank $r$ matrix $A \in \mathcal{S}_+^n$ by first randomly generating a $C \in \mathbb{R}^{n \times r}$ with entries individually drawn from a $\mathcal{N}(0,1)$ distribution and forming $A$ as $A = CC^T$. The *block diagonal* mask is formed from $5 \times 5$ principal submatrices running down the diagonal, each principal submatrix overlapping the one to its upper left. Such a mask might be encountered in practice if we obtain pairwise measurements from small sets of variables sequentially. The $l^{th}$ principal submatrix of the *full columns* mask is formed by sampling all pairs of entires, $(i, j)$ indexed by $i, j \in \{1, 2, 3, 4, l+4\}$ and might be encountered when obtaining pairwise measurements between sets of variables, where some small number of variables is present in all sets. The *random* mask is formed from principal submatrices randomly generated to conform to assumptions $A1$ through $A3$ and demonstrates that masks with non-obvious structure in the underlying principal submatrices can conform to assumptions $A1$ through $A3$. Algorithm 1 correctly recovers the true matrix from all three masks.

In panel Fig. 2B, we modify these three types of masks so that $\min_l |\iota_l|$, the minimal overlap of a principal submatrix with those ordered before it, is 10 for each and attempt to reconstruct random matrices of size $55 \times 55$ and increasing rank. Corollaries $7-9$ in the appendix, which can be derived from Theorem 1 above, can be applied to these scenarios to establish the necessity that $\min_l |\iota_l|$ be greater than $r$ for a rank $r$ matrix. As predicted, for all masks recovery is successful for all matrices of rank 10 or less and unsuccessful for matrices of greater rank. In Fig. 2C, we show this is not unique to masks with minimal overlap of 10. Here we generate block diagonal masks with minimal overlap between the principal submatrices varying between 0 and 54. For each overlap value, we then attempt to recover matrices of rank 1 through $o + 1$, where $o$ is the minimal overlap value. To guard against false positives, we randomly generated 10 matrices of a specified rank for each mask and only indicated success in black if matrix completion was successful in all cases. As predicted by theory, matrix completion failed exactly when the rank of the underlying matrix exceeded the minimal overlap value of the mask. Identical results were obtained for the full column and random masks.

We provide evidence the dependence on $\epsilon$ in Theorem 4 should be linear in Fig. 3. We generate random $55 \times 55$ matrices of rank 1 through 10. Matrices were generated as in the noiseless scenario and normalized to have a Frobenius norm of 1. We use a block diagonal mask with $25 \times 25$ blocks and

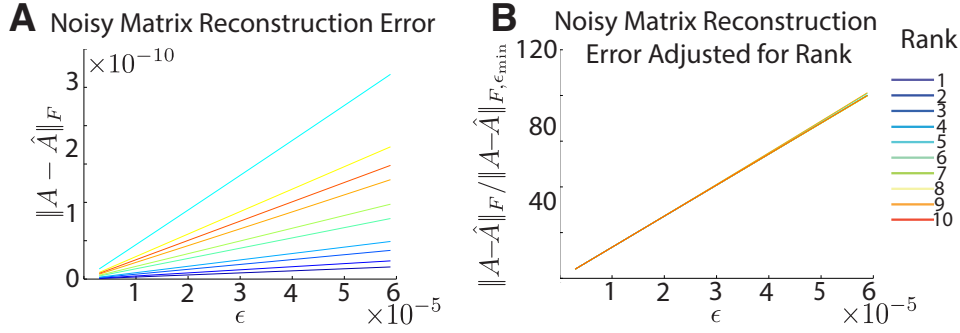

Figure 3: Noisy simulation results. (A) Reconstruction error with increasing amounts of noise applied to the original matrix. (B) Traces in panel (A), each divided by its value at $\epsilon = \epsilon_{\min}$.

an overlap of 15 and randomly generate SPSD noise, scaled so that $\|A_l - \tilde{A}_l\| = \epsilon$ for each principal submatrix. We sweep through a range of $\epsilon \in [\epsilon_{\min}, \epsilon_{\max}]$ for a $\epsilon_{\min} > 0$ and a $\epsilon_{\max}$ determined by the matrix with the tightest constraint on $\epsilon$ in theorem 4. Fig. 3A shows that reconstruction error generally increases with $\epsilon$ and the rank of the matrix to be recovered. To better visualize the dependence on $\epsilon$, in Fig. 3B, we plot $\|A - \hat{A}\|_F / \|A - \hat{A}\|_{F, \epsilon_{\min}}$, where $\|A - \hat{A}\|_{F, \epsilon_{\min}}$ indicates the reconstruction error obtained with $\epsilon = \epsilon_{\min}$. All of the lines coincide, suggesting a linear dependence on $\epsilon$.

## 7 Discussion

In this work we present an algorithm for the recovery of a SPSD matrix from a deterministic sampling of its principal submatrices. We establish sufficient conditions for our algorithm to exactly recover a SPSD matrix and present a set of necessity results demonstrating that our stated conditions can be quite useful for determining when matrix recovery is possible by any method. We also show that our algorithm recovers matrices obscured by noise with increasing fidelity as the magnitude of noise goes to zero. Our algorithm incorporates no tuning parameters and can be computationally light, as the majority of computations concern potentially small principal submatrices of the original matrix. Implementations of the algorithm, which estimate each $\hat{C}_l$ in parallel, are also easy to construct. Additionally, our results can be generalized when the principal submatrices our method uses for reconstruction are themselves not fully observed. In this case, existing matrix recovery techniques can be used to estimate each complete underlying principal submatrix with some bounded error. Our algorithm can then reconstruct the full matrix from these estimated principal submatrices.

An open question is the computational complexity of finding a set of principal submatrices which satisfy conditions $A1$ through $A4$. However, in many practical situations there is an obvious set of principal submatrices and ordering which satisfy these conditions. For example, in the neuroscience application described in the introduction, a set of recording probes are independently movable and each probe records from a given number of neurons in the brain. Each configuration of the probes corresponds to a block of simultaneously recorded neurons, and by moving the probes one at a time, blocks with overlapping variables can be constructed. When learning a low rank covariance structure for this data, the overlapping blocks of variables naturally define observed blocks of a low rank covariance matrix to use in algorithm 1.

**Acknowledgements**

This work was supported by an NDSEG fellowship, NIH grant T90 DA022762, NIH grant R90 DA023426-06 and by the Craig H. Nielsen Foundation. We thank Martin Azizyan, Geoff Gordon, Akshay Krishnamurthy and Aarti Singh for their helpful discussions and Rob Kass for his guidance.

## Footnotes

[1] Throughout this work we will use MATLAB indexing notation, so $C(\rho_i, :)$ is the submatrix of $C$ made up of the rows indexed by the ordered set $\rho_i$.

[2]We use the notation, $[n]$ to indicate the set $\{1, \ldots, n\}$.

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
