[Supplementary Material]

# Appendix for Deterministic Symmetric Positive Semidefinite Matrix Completion

**William Bishop**[1,2], **Byron Yu**[2,3,4]
[1]Machine Learning, [2]Center for the Neural Basis of Cognition,
[3]Biomedical Engineering, [4]Electrical and Computer Engineering
Carnegie Mellon University
{wbishop, byronyu}@cmu.edu

## A   Proofs of the Main Results

### A.1   Proof of Theorem 1

*Proof.* The theorem is a restatement of lemma 6, which we state and prove in the technical results section below. For clarity, here we formally establish the correspondence between the theorem and lemma 6.

Let $\Omega_1 = \rho_1 \times \rho_1$ and $\Omega_2 = \rho_2 \times \rho_2$ be the two index sets in the theorem. By assumption we have $\rho_1 \times \rho_1 \cup \rho_2 \times \rho_2 = \Omega$ and $\Omega \neq [n] \times [n]$. If $A1$ is not met, then $\rho_1 \cup \rho_2 \neq [n]$, and from lemma 6 we can conclude recovery of $A$ is impossible.

If $\rho_1 \cup \rho_2 = [n]$, but $A2$ is not met then $|\rho_1 \cap \rho_2| < r$ so it must be that **rank**$\{A(\iota_2, \iota_2)\} < r$, and the final condition of the lemma is not met, so matrix recovery is impossible. Finally, if $A3$ is not met, then again **rank**$\{A(\iota_2, \iota_2)\} < r$ and we can conclude matrix recovery is impossible.

$\square$

### A.2   Proof of Theorem 2

*Proof.* Our proof technique will be to show that at the end of step 2 of the algorithm, the algorithm has learned a $\hat{C}$ such that $\hat{A} = \hat{C}\hat{C}^T = A$ in step 3 of the algorithm.

We establish the result by induction, using $k = 1$ as the base case. If $k = 1$, algorithm 1 consists of running step 1 followed immediately by step 3. The matrix $A(\rho_{\tau_1}, \rho_{\tau_1})$ is a principal submatrix of $A$ by assumption $A2$ and must therefore be SPSD. Therefore, we can decompose it with a real eigendecomposition, and $\hat{C}(\rho_{\tau_1}, :) = E_{\tau_1}\Lambda_{\tau_1}^{1/2}$ in step 1 of the algorithm, so it must be that $\hat{C}(\rho_{\tau_1}, :) \in \mathbb{R}^{|\rho_{\tau_1}| \times r}$ and

$$\hat{A}(\rho_{\tau_1}, \rho_{\tau_1}) = \hat{C}(\rho_{\tau_1}, :)\hat{C}(\rho_{\tau_1}, :)^T = E_{\tau_1}\Lambda_{\tau_1}E_{\tau_1}^T = A(\rho_{\tau_1}, \rho_{\tau_1}).$$

If $k > 1$, there are $k - 1$ iterations of step 2 between step 1 and step 3 of the algorithm. Let $l \in \{2, \ldots, k\}$ indicate[1] what iteration the algorithm is on in step 2. Define $\mathcal{I}_l = \cup_{i=1}^{l}\rho_{\tau_i}$. Assume at the start of the iteration of step 2 for a given $l$, the algorithm has previously assigned values to $\hat{C}(\mathcal{I}_{l-1}, :) \in \mathbb{R}^{|\mathcal{I}_{l-1}| \times r}$ so that

$$\hat{C}(\mathcal{I}_{l-1}, :)\hat{C}(\mathcal{I}_{l-1}, :)^T = \hat{A}(\mathcal{I}_{l-1}, \mathcal{I}_{l-1}) = A(\mathcal{I}_{l-1}, \mathcal{I}_{l-1}).$$

Above we have shown this is true for $l = 2$. For part 2a of each iteration, there is again by assumption no noise, so $\tilde{A}(\rho_{\tau_l}, \rho_{\tau_l}) = A(\rho_{\tau_l}, \rho_{\tau_l})$. $A(\rho_{\tau_l}, \rho_{\tau_l})$ again indexes a principal submatrix of $A$ by

assumption $A2$, so its eigendecomposition with real eigenvalues exists and we again have $\hat{C}_l = E_{\tau_l} \Lambda_{\tau_l}^{1/2}$, $\hat{C}_l \in \mathbb{R}^{|\rho_{\tau_l}| \times r}$ for $\hat{C}_l$ in step 2a, so

$$\hat{C}_l \hat{C}_l^T = E_{\tau_l} \Lambda_{\tau_l} E_{\tau_l}^T = A(\rho_l, \rho_l).$$

We now use lemma 5 to establish that after steps b and c, we will have a $\hat{C}(\mathcal{I}_l, :) \in \mathbb{R}^{|\mathcal{I}_l| \times r}$ such that $\hat{C}(\mathcal{I}_l, :)\hat{C}(\mathcal{I}_l, :)^T = A(\mathcal{I}_l, , \mathcal{I}_l, )$.

To establish that this lemma applies, we need to first establish the correspondence between the variables of algorithm 1 and those used in the statement of the lemma. Specifically, $A(\mathcal{I}_l, \mathcal{I}_l)$ is the matrix to be reconstructed from two of it's principal submatrices $A(\mathcal{I}_{l-1}, \mathcal{I}_{l-1})$ and $A(\rho_{\tau_l}, \rho_{\tau_l})$. We can verify that as required by the lemma $A(\rho_{\tau_l} \cup \mathcal{I}_{l-1}, \rho_{\tau_l} \cup \mathcal{I}_{l-1}) = A(\mathcal{I}_l, \mathcal{I}_l)$. For these two index sets, we have also already established that the algorithm's variables $\hat{C}(\mathcal{I}_{l-1}, :)$ and $\hat{C}_l$ are such that $\hat{C}(\mathcal{I}_{l-1}, :) \in \mathbb{R}^{|\mathcal{I}_{l-1}| \times r}$, $\hat{C}_l \in \mathbb{R}^{|\rho_{\tau_l}| \times r}$, $\hat{C}(\mathcal{I}_{l-1}, :)\hat{C}(\mathcal{I}_{l-1}, :)^T = A(\mathcal{I}_{l-1}, \mathcal{I}_{l-1})$ and $\hat{C}_l \hat{C}_l^T = A(\rho_{\tau_l}, \rho_{\tau_l})$.

Continuing to establish the correspondence between variables in the algorithm and the lemma, we note that the algorithm's variable $\iota_l = \rho_{\tau_l} \cap \left(\cup_{j=1,\ldots,l-1}\rho_{\tau_j}\right) = \rho_{\tau_l} \cap \mathcal{I}_{l-1}$. Further, it can be verified that the variables in the algorithm $\iota_l$ and $\phi_l$ index the rows of $\hat{C}$ and $\hat{C}_l$ such that $\hat{C}(\iota_l, :)\hat{C}(\iota_l, :)^T = A(\iota_l, \iota_l) = \hat{C}_l(\phi_l, :)\hat{C}_l(\phi_l, :)^T$, and it can also be verified that algorithm's variable $\eta_l$ is defined so that $A(\rho_{\tau_l} \setminus \iota_l, \rho_{\tau_l} \setminus \iota_l) = \hat{C}_l(\eta_l, :)\hat{C}_l(\eta_l, :)^T$.

By assumptions $A3$ and $A4$, **rank**$\{A(\iota_l, \iota_l)\} = r$, so it then follows from the lemma that there must be a unique orthogonal $W$ such that $\hat{C}(\iota_l, :) = \hat{C}_l(\phi_l, :)W$. However, if $W$ is unique and establishes the equality, this must be the $\hat{W}_l$ learned in part b of step 2 of the algorithm. Further, we just established the correspondence between the variables in the algorithm and that of the lemma, so in part c of the step 2, we can conclude the algorithm produces a $\hat{C}(\mathcal{I}_l, :)$ such that $A(\mathcal{I}_l, \mathcal{I}_l) = \hat{C}(\mathcal{I}_l, :)\hat{C}(\mathcal{I}_l, :)^T$, which completes the inductive part of the proof.

All that remains of the proof is to show after processing all of the principal submatrices indexed by $\Omega_1, \ldots, \Omega_k$, in steps 1 and 2 of the algorithm, step 3 will exactly recover $A$. If $k = 1$, the algorithm consists of step 1 followed immediately by step 3. In step 1 we have shown that it will assign values to the rows of $\hat{C}(\mathcal{I}_1, :)$ such that $\hat{C}(\mathcal{I}_1, :)\hat{C}(\mathcal{I}_1, :)^T = A(\mathcal{I}_1, \mathcal{I}_1)$. For $k > 2$, the algorithm will perform $k - 1$ iterations of step 2 before step 3. By the inductive part of the proof just finished, we can conclude after these $k - 1$ iterations, the algorithm will have assigned values to the rows of $\hat{C}(\mathcal{I}_k, :)$ such that $\hat{C}(\mathcal{I}_k, :)\hat{C}(\mathcal{I}_k, :)^T = A(\mathcal{I}_k, \mathcal{I}_k)$. However, by assumptions $A1$ and $A2$, $\mathcal{I}_k = \cup_{l=1}^k \rho_{\tau_l} = [n]$, showing that all the rows of $\hat{C}$ have been learned and therefore in step 3 of the algorithm $\hat{A} = \hat{C}\hat{C}^T = A$.

$\square$

### A.3 Proof of Theorem 3

*Proof.* The proof is by construction. First, consider the case when $A$ is full rank and $r = n$. In this case, we simply sample the full matrix and $A1 - A3$ trivially hold. Further, $|\Omega| = n^2 = nr < n(2r + 1)$.

Now, consider the case when $r < n$. Let $\Omega$ index the set of $k = n - r$ principal submatrices running down the diagonal of $A$, such that for $l \in \{1, \ldots, k\}$, $\rho_l = \{1, \ldots, r+1\} + \{l-1\}$.[2] We can verify that

$$\boldsymbol{\rho}\{\Omega\} = \boldsymbol{\rho}\{\cup_{l=1}^k \Omega_l\} = \cup_{l=1}^k \rho_l = [n],$$

meeting $A1$. Further, by construction each $\Omega_l$ satisfies $A2$, and if we let $\tau_1, \ldots, \tau_k = 1 \ldots, k$, we have that for $i \geq 2$

$$\left|\rho_{\tau_i} \cap \left(\cup_{j \in \{1,\ldots,i-1\}} \rho_{\tau_j}\right)\right| = \left|\rho_{\tau_i} \cap \rho_{\tau_{i-1}}\right| = r,$$

establishing that $A3$ holds as well. Having established that $A1 - A3$ hold for the proposed $\Omega$, we count the number of indexed entries. The set $\Omega_1$ indexes $(r+1)^2$ entries, and each further $\Omega_k$ for $k \geq 2$ indexes an additional $2r+1$ entries not index by earlier $\Omega_k$. We can formally count these entries as

$$
\begin{aligned}
|\Omega| &= |\Omega_1| + \sum_{j=2}^{n-r} \left|\Omega_j \setminus \left(\Omega_j \cap \left[\cup_{l=1}^{j-1} \Omega_l\right]\right)\right| \\
&= (r+1)^2 + \sum_{j=2}^{n-r} (2r+1) \\
&= (r+1)^2 + (n-r-1)(2r+1) \\
&= 2nr + n - r^2 - r \\
&\leq n(2r+1).
\end{aligned}
$$

$\square$

## A.4   Proof of Theorem 4

*Proof.* For the rank $r$ matrix $A \in \mathcal{S}_+^n$, fix $C \in \mathbb{R}^{n \times r}$ such that $A = CC^T$. By lemma 1, such a $C$ must exist. Now, define $C_l = C(\rho_l, :)$. By assumption $A2$, each $A_l \in \mathcal{S}_+^{n_l}$, so $A_l = C_l C_l^T$ and by assumption $A4$, $\mathbf{rank}\{A_l\} = r$ for all $l$, as $\mathbf{rank}\{A(\iota_l, \iota_l)\} = r$ for all $l \geq 2$.

Additionally, by assumption $A6$, $\tilde{A}_l \in \mathcal{S}_+^{n_l}$, by assumption $A5$, $\mathbf{rank}\left\{\tilde{A}_l\right\} \geq r$, and $\|A_l - \tilde{A}_l\|_F \leq \epsilon < 1$ for all $l$. Further, we have $\min\{\min_{i \in [r-1]}, |\lambda_{l,i} - \lambda_{l,i+1}|, \lambda_{l,r}\} \geq \delta > 0$ for $l \in \{1, \ldots, k\}$, and we also have $\epsilon < \delta/2$. Then by lemma 15, if each $\hat{C}_l$ is formed according to steps 1 and 2a of algorithm[3] 1

$$\min_{W:WW^T=I} \|C_l - \hat{C}_l W\|_F \leq L\sqrt{r}\epsilon,$$

for $L = \sqrt{1 + \frac{16\zeta}{\delta^2} + \frac{8\sqrt{2}\zeta^{1/2}}{\delta^{3/2}}}$, where we have used the assumption $\zeta \geq \lambda_{l,1}$ for all $l$.

Forming $\hat{C}$ from the $\hat{C}_l$ matrices in steps 1 and 2 of algorithm 1 is equivalent to forming $\hat{C}$ from the same $\hat{C}_l$ matrices with algorithm 2 in corollary 19 according to the ordering given by $\tau_1, \ldots, \tau_k$. Thus, we can use 19 to bound the error on $\min_{W:WW^T=I} \left\|C - \hat{C}W\right\|_F$ at the end of step 2 of algorithm 1.

Corollary 19 requires that $\|C(\rho_l, :)\| \leq b$ for some $b \geq 0$ and $\min_{W:WW^T=I} \left\|C_l - \hat{C}_l W\right\|_F \leq \epsilon^*/b \leq 1$ for some $\epsilon^* \geq 0$ for all $l$.

By assumption $b \geq \max_{l \in [k]} \|C_l^*\|_F$ for some $C_l^*$ such that $A_l = C_l^* C_l^{*T}$. However, we can show that there exists an orthogonal $O$ such that $C_l = C_l^* O$ for any $C_l$ and $C_l^*$ such that $C_l^* C_l^{*T} = A_l = C_l C_l^T$. The Frobenius norm is a unitarily invariant norm, so if $b \geq \max_{l \in [k]} \|C_l^*\|_F$, then $b \geq \max_{l \in [k]} \|C_l\|_F$, as defined above, and it must be that $\|C(\rho_l, :)\|_F \leq b$ for all $l$. Defining $\epsilon^* = L\sqrt{r}\epsilon$, and noting that our theorem requires $\sqrt{r}\epsilon < b$, we can verify

$$\frac{\epsilon^*}{b} = \frac{L\sqrt{r\epsilon}}{b} \leq \frac{\sqrt{r\epsilon}}{b} < 1,$$

satisfying the requirements of the lemma.

Corollary 19 also requires that $C(\iota_l,:)$ and $\hat{C}_l(\phi_l,:)^T\hat{C}(\iota_l,:)$ each are of rank $r$ for $l \in 2,\ldots,k$. However, this assumption is also met. To see this, note that by assumptions $A3$ and $A4$ $A(\iota_l,\iota_l) = C(\iota_l,:)C(\iota_l,:)^T$ is of rank $r$ for all $l \geq 2$ so $C(\iota_l,:)$ must be of rank $r$ for all $l \geq 2$. The requirement that $\hat{C}_l(\phi_l,:)^T\hat{C}(\iota_l,:)$ is of rank $r$ for all $l \geq 2$ is satisfied by the assumptions of the theorem. Finally, corollary 19 also requires that $\cup_{l=1}^{k}\rho_l = [n]$, which is met by assumption $A1$.

Therefore, we can use corollary 19 to bound $\min_{W:WW^T=I}\left\|C - \hat{C}W\right\|_F$

as

$$\min_{W:WW^T=I}\left\|C - \hat{C}W\right\|_F \leq [4 + 12/v]^{k-1}\epsilon^*$$
$$= [4 + 12/v]^{k-1}L\sqrt{r\epsilon}$$

when $v \leq \sigma_r^2(C(\iota_l,:))/b^2$ for all $l$. Note that $\lambda_r(A(\iota_l,\iota_l)) = \sigma_r^2(C(\iota_l,:))$, so this condition on $v$ is equivalent to $v \leq \lambda_r(A(\iota_l,\iota_l))/b^2$ for all $l$.

To obtain a bound on $\left\|A - \hat{A}\right\|_F$, we apply lemma 11, to find

$$\left\|A - \hat{A}\right\|_F = \left\|CC^T - \hat{C}\hat{C}^T\right\|$$
$$\leq 2\left\|C\right\|_F\left([4 + 12/v]^{k-1}L\sqrt{r\epsilon}\right) + \left([4 + 12/v]^{k-1}L\sqrt{r\epsilon}\right)^2$$
$$= 2\left\|C\right\|_F\left([4 + 12/v]^{k-1}L\sqrt{r\epsilon}\right) + [4 + 12/v]^{2k-2}L^2r\epsilon,$$

defining $G = 4 + 12/v$, we have

$$\left\|A - \hat{A}\right\|_F \leq 2G^{k-1}L\left\|C\right\|_F\sqrt{r\epsilon} + G^{2k-2}L^2r\epsilon. \tag{1}$$

Note that the statement of the main theorem in the body of the paper allows $C$ to be any arbitrary $n$ by $r$ matrix such that $A = CC^T$. We note that indeed any such $C$ will do as if $A = CC^T = C'C'^T$ for some $C' \in \mathbb{R}^{n\times r}$, there is an orthogonal transformation relating $C$ and $C'$. The Frobenius norm is a unitarily invariant norm, so $\|C\|_F = \|C'\|_F$, which completes the proof.

$\square$

## B    Technical Results

**Lemma 1.** *For any $A \in \mathcal{S}_+^n$ such that **rank**$\{A\} \leq r$, there exists a matrix $C \in \mathbb{R}^{n\times r}$ such that $A = CC^T$.*

*Proof.* By the spectral theorem for real symmetric matrices, any rank $p$ symmetric, real matrix can be decomposed as

$$A = E\Lambda E^T,$$

for some diagonal $\Lambda \in \mathbb{R}^{p\times p}$ containing the $p$ non-zero eigenvalues of $A$ and $E \in \mathbb{R}^{n\times p}$, containing the corresponding eigenvectors. Further, the $p$ non-zero eigenvalues of $A$ must all be

greater than 0 as $A$ is $PSD$. Define $C = E\Sigma^{1/2}$. By construction $C \in \mathbb{R}^{n \times p}$ and we have $CC^T = E\Sigma^{1/2}(E\Sigma^{1/2})^T = E\Sigma E^T = A$. If $p = r$, this establishes the existence of a $C \in \mathbb{R}^{n \times r}$ such that $A = CC^T$. If $p < r$, a $C \in \mathbb{R}^{n \times r}$ such that $A = CC^T$ can be constructed by adding $r - p$ columns of zeros to the matrix $E\Sigma^{1/2}$. $\qquad \square$

**Lemma 2.** *Let $C \in \mathbb{R}^{n \times p}$ be a rank $r$ matrix. For the orthogonal matrix $W \in \mathbb{R}^{p \times p}$, consider the equality $C = CW$. If $r < p$, then $W$ is non-unique. Further, $W = I$ if and only if $r = p$.*

*Proof.* We begin with the case $r = p$. First, if $W = I$, then it immediately follows that $C = CI$. Now, assume $r = p$. If $r = p$, it must be that $n \geq p$. We write the singular value decomposition of $C$ as $C = U\Sigma V^T$ for some $U \in \mathbb{R}^{n \times p}$ with orthonormal columns, orthogonal $V \in \mathbb{R}^{p \times p}$ and diagonal $\Sigma \in \mathbb{R}^{p \times p}$ with the $p$ non-zero singular values of $C$ along its diagonal. We now solve for $W$ in the equation $C = CW$ to show that uniquely $W = I$. Specifically, we have

$$CW = C$$
$$U\Sigma V^T W = U\Sigma V^T$$
$$(V\Sigma^{-1}U^T)U\Sigma V^T W = (V\Sigma^{-1}U^T)U\Sigma V^T$$
$$W = I.$$

We now consider $r < p$. Denote the singular value decomposition of $C$ as $C = U\Sigma V^T$. We now need to consider the possibility that $n < p$, so define $U \in \mathbb{R}^{n \times n}$ and $V \in \mathbb{R}^{p \times p}$ to be orthogonal matrices and let $\Sigma \in \mathbb{R}^{n \times p}$ be a matrix with the singular values of $C$ in the appropriate locations. Specifically, the diagonal of $\Sigma$ will contain the $r$ singular values of $C$ and $p - r$ zero values along its diagonal. We again consider solutions to the equation $U\Sigma V^T W = U\Sigma V^T$. We can left multiply by $U^T$ to obtain $\Sigma V^T W = \Sigma V^T$. Let $W = VO$ for some orthogonal $O \in \mathbb{R}^{p \times p}$. Let $O(i,:) = V(:,i)^T$ if $\Sigma(i,i) \neq 0$. Then it can be verified that $\Sigma V^T = \Sigma V^T W$, regardless of the values of the remaining rows of $O$. Furter, the remaining rows of $O$ can be chosen arbitrarily as long as $O$ remains orthogonal. In the case of a single missing row, the missing row is only determined up to multiplication by 1 or $-1$. In the case of more than one missing row, there is greater flexibility. When $O$ is non-unique, the product $W = VO$ will be non-unique, completing the proof.

$\qquad \square$

**Corollary 3.** *Let $C_1, C_2 \in \mathbb{R}^{n \times r}$ be matrices such that $C_1 = C_2W$ for some orthogonal $W \in \mathbb{R}^{r \times r}$. If the rank of $C_1$ is $r$, then $W$ is unique.*

*Proof.* The proof is by contradiction. First, assume there are two distinct orthogonal matrices $W_1$ and $W_2$, such that $C_1 = C_2W_1$ and $C_1 = C_2W_2$. Then it must be that

$$C_2 = C_2 W_1 W_2^T. \tag{2}$$

Next, note that if $W_1$ and $W_2$ are two distinct orthornormal matrices, $W_1 W_2^T \neq I$. However, by assumption $C_2$ is of rank $r$ and by lemma 2 (2) holds if and only if $W_1 W_2^T = I$, and we have achieved a contradiction.

$\qquad \square$

**Lemma 4.** *For the rank $r$ matrix $A \in \mathcal{S}_+^n$ define $C_1, C_2 \in \mathbb{R}^{n \times r}$ such that $A = C_1 C_1^T$ and $A = C_2 C_2^T$. Then there exists a unique orthogonal $W \in \mathbb{R}^{r \times r}$ such that $C_1 = C_2 W$.*

*Proof.* First note that by assumption **rank**$\{A\} = r$, so $C_1$ and $C_2$ must also be of rank $r$. Further, it must be that **rank**$\{A\} \leq n$, so $r \leq n$. Denote the singular value decomposition of $C_1$ as $C_1 = U_1 \Sigma_1 V_1^T$, for $U_1 \in \mathbb{R}^{n \times r}$ with orthonormal columns, diagonal $r \times r$ matrix $\Sigma_1$, with the non-zero singular values of $C_1$ along its diagonal, and orthogonal $V_1 \in \mathbb{R}^{r \times r}$. Use a similar notation for the SVD of $C_2$. With these preliminaries out of the way, we will first establish that $W$ exists and is orthogonal and then call on corollary 3 to establish its uniqueness.

First, we establish that $W$ exists by noting that we can solve for it. All of the singular values of $C_1$ and $C_2$ must be strictly greater than 0, as $C_1$ and $C_2$ are of rank $r$. Therefore, $\Sigma_1^{-1}$ and $\Sigma_2^{-1}$ are well defined. Thus, we can solve for $W$ as $W = V_2 \Sigma_2^{-1} U_2^T U_1 \Sigma_1 V_1^T$.

To establish that $W$ must be orthogonal let $W$ be a matrix such that $C_1 = C_2 W$ and note that we have

$$A = C_1 C_1^T = C_2 W W^T C_2^T = C_2 C_2^T = A,$$

so clearly $C_2 C_2^T = C_2 W W^T C_2^T$.

Using our notation for the SVD of $C2$, we can then write

$$U_2 \Sigma_2 V_2^T I V_2 \Sigma_2 U_2^T = U_2 \Sigma_2 V_2^T W W^T V_2 \Sigma_2 U_2^T. \tag{3}$$

Noting again that $\Sigma_2^{-1}$ is well defined, we can then left multiple both sides of (3) by $V_2 \Sigma_2^{-1} U_2^T$ and right multiply by $U_2 \Sigma_2^{-1} V_2^T$ to find,

$$I = W W^T,$$

establishing that $W$ must be orthogonal. Having established that there exists an orthogonal $W$ such that $C_1 = C_2 W$, that $C_1$ and $C_2$ are of rank $r$, the uniqueness of $W$ follows directly from corollary 3. $\qquad\square$

**Lemma 5.** *Consider the rank $r$ matrix $A \in \mathcal{S}_+^n$. Define the index sets $\mathcal{I}_1$ and $\mathcal{I}_2$ such that $A(\mathcal{I}_1, \mathcal{I}_1)$ and $A(\mathcal{I}_2, \mathcal{I}_2)$ are submatrices of $A$ and $A(\mathcal{I}_1 \cup \mathcal{I}_2, \mathcal{I}_1 \cup \mathcal{I}_2) = A$. Let $\iota = \mathcal{I}_1 \cap \mathcal{I}_2$. Let $n_1 = |\mathcal{I}_1|$, and $n_2 = |\mathcal{I}_2|$.*

*Define $C_1 \in \mathbb{R}^{n_1 \times r}$ and $C_2 \in \mathbb{R}^{n_2 \times r}$ such that $A(\mathcal{I}_1, \mathcal{I}_1) = C_1 C_1^T$ and $A(\mathcal{I}_2, \mathcal{I}_2) = C_2 C_2^T$.*

*Further, let $\phi_1$ and $\phi_2$ index the rows of $C_1$ and $C_2$, respectively, such that $C_1(\phi_1, :) C_1(\phi_1, :)^T = A(\iota, \iota) = C_2(\phi_2, :) C_2(\phi_2, :)^T$. Finally, let $\eta_2$ index the rows of $C_2$ such that $A(\mathcal{I}_2 \setminus \iota, \mathcal{I}_2 \setminus \iota) = C_2(\eta_2, :) C_2(\eta_2, :)^T$. Then if the rank of $A(\iota, \iota)$ is equal to $r$, there exists a unique orthogonal*

$$W : C_1(\phi_1, :) = C_2(\phi_2, :) W \tag{4}$$

*such that $A = C' C'^T$ when $C' \in \mathbb{R}^{n \times r}$ is formed as*

$$C'(\mathcal{I}_1, :) = C_1$$
$$C'(\mathcal{I}_2 \setminus \iota, :) = C_2(\eta_2, :) W.$$

*Proof.* Note that

$$C_1(\phi_1, :) C_1(\phi_1, :)^T = C_2(\phi_2, :) C_2(\phi_2, :)^T = A(\iota, \iota).$$

Further, by assumption $A(\iota, \iota)$ is of rank $r$, so $C_1(\phi_1, :)$ and $C_2(\phi_2, :)$ must be as well. Then the existence, ortogonality and uniqueness of $W$ follow from lemma 4.

Next, we establish that $C' C'^T = A$. First note that

$$C'(\mathcal{I}_1, :) C'(\mathcal{I}_1, :)^T = C_1 C_1^T = A(\mathcal{I}_1, \mathcal{I}_1),$$

and

$$C'(\mathcal{I}_2 \setminus \iota, :)C'(\mathcal{I}_2 \setminus \iota, :)^T = C_2(\eta_2, :)WW^T C_2(\eta_2, :)^T$$
$$= C_2(\eta_2, :)C_2(\eta_2, :)^T$$
$$= A(\mathcal{I}_2 \setminus \iota, \mathcal{I}_2 \setminus \iota).$$

All that remains is to show that $C'(\mathcal{I}_2 \setminus \iota, :)C'(\mathcal{I}_1, :)^T = A(\mathcal{I}_2 \setminus \iota, \mathcal{I}_1)$. We do this by reference to an arbitrary matrix $D \in \mathbb{R}^{n \times r}$ such that $A = DD^T$. It must be that $A(\mathcal{I}_1, \mathcal{I}_1) = D(\mathcal{I}_1, :)D(\mathcal{I}_1, :)^T$ and $A(\mathcal{I}_2, \mathcal{I}_2) = D(\mathcal{I}_2, :)D(\mathcal{I}_2, :)^T$. Additionally, we also know that $A(\mathcal{I}_1, \mathcal{I}_1) = C_1 C_1^T$ and $A(\mathcal{I}_2, \mathcal{I}_2) = C_2 C_2^T$.

Further, the rank $r$ matrix $A(\iota, \iota)$ is a submatrix of $A(\mathcal{I}_1, \mathcal{I}_1)$ and $A(\mathcal{I}_2, \mathcal{I}_2)$ so $A(\mathcal{I}_1, \mathcal{I}_1)$ and $A(\mathcal{I}_2, \mathcal{I}_2)$ must also be of rank $r$, from which it follows that $D(\mathcal{I}_1, :)$, $D(\mathcal{I}_2, :)$, $C_1$ and $C_2$ are also of rank $r$.

From lemma 4 it then follows that there exist unique orthogonal $O_1$ and $O_2$ such that $D(\mathcal{I}_1, :) = C_1 O_1$ and $D(\mathcal{I}_2, :) = C_2 O_2$. Further, we have

$$C_1(\phi_1, :)C_1(\phi_1, :)^T = C_2(\phi_2, :)WC_1(\phi_1, :)^T$$
$$= D(\iota, :)O_2^T WO_1 D(\iota, :)^T$$
$$= A(\iota, \iota),$$

which establishes that $D(\iota, :)O_2^T WO_1 D(\iota, :)^T = A(\iota, \iota)$. Further, $D(\iota, :)$ is a rank $r$ matrix of size $|\iota|$ by $r$. Therefore, for the matrix $M$ it must be that $D(\iota, :)MD(\iota, :)^T = A(\iota, \iota) = D(\iota, :)D(\iota, :)^T$ if and only if $M = I$. Therefore, $D(\iota, :)O_2^T WO_1 D(\iota, :)^T = A(\iota, \iota)$ if and only if $O_2 WO_1^T = I$.

Now, consider, $C'(\mathcal{I}_2 \setminus \iota, :)C'(\mathcal{I}_1, :)^T$. We can write

$$C'(\mathcal{I}_2 \setminus \iota, :)C'(\mathcal{I}_1, :)^T = C_2(\eta_2, :)WC_1^T$$
$$= D(\mathcal{I}_2 \setminus \iota, :)O_2^T WO_1 D(\mathcal{I}_1, :)^T$$
$$= D(\mathcal{I}_2 \setminus \iota, :)ID(\mathcal{I}_1, :)^T$$
$$= A(\mathcal{I}_2 \setminus \iota, \mathcal{I}_1).$$

Having shown that $A(\mathcal{I}_1, \mathcal{I}_1) = C'(\mathcal{I}_1, :)C'(\mathcal{I}_1, :)^T$, $A(\mathcal{I}_2 \setminus \iota, \mathcal{I}_2 \setminus \iota) = C'(\mathcal{I}_2 \setminus \iota, :)C'(\mathcal{I}_2 \setminus \iota, :)^T$ and $A(\mathcal{I}_2 \setminus \iota, \mathcal{I}_1) = C'(\mathcal{I}_2 \setminus \iota, :)C'(\mathcal{I}_1, :)^T$, we can conclude that $A(\mathcal{I}_1 \cup \mathcal{I}_2, \mathcal{I}_1 \cup \mathcal{I}_2) = C'C'^T$. Further, $A(\mathcal{I}_1 \cup \mathcal{I}_2, \mathcal{I}_1 \cup \mathcal{I}_2)$ must be the entirety of $A$ as by assumption $A(\mathcal{I}_1 \cup \mathcal{I}_2, \mathcal{I}_1 \cup \mathcal{I}_2) = A$, which completes the proof.

$\square$

**Lemma 6.** *Let $A \in \mathcal{S}_+^n$ be a rank $r$ matrix. Define the index sets $\mathcal{I}_1 \subset [n]$ and $\mathcal{I}_2 \subset [n]$ and let $\iota = \mathcal{I}_1 \cap \mathcal{I}_2$. Let $\Omega$ denote the set of observed entries of $A$ and assume $\Omega = \mathcal{I}_1 \times \mathcal{I}_1 \cup \mathcal{I}_2 \times \mathcal{I}_2$ and $\Omega \neq [n] \times [n]$. Then it is necessary that $\mathcal{I}_1 \cup \mathcal{I}_2 = [n]$ and **rank**$\{A(\iota, \iota)\} = r$, where it is understood that **rank**$\{A(\iota, \iota)\} = 0$ if $\iota = \emptyset$, for there to be a unique rank $r$ completion of $A$.*

*Proof.* First, if $\mathcal{I}_1 \cup \mathcal{I}_2 \neq [n]$, the set $\Omega$ fails to index a complete row and column of $A$, which we compactly express as $\{(i,j),(j,i) : j \in [n]\} \cap \Omega = \emptyset$ for some $i \in [n]$. The matrix $A$ is PSD, and by lemma 1 we can decompose $A$ as $A = CC^T$ for some $C \in \mathbb{R}^{n \times r}$. For any such $C$, we can write the individual entries of $A$ as $A(i,j) = C(i, :)C(j, :)^T$ for all $i, j \in [n]$. For row $i$ of any such $C$, the set of entries $\{A_{ij}, A_{ji} : j \in [n]\}$ then provide the only set of equality constraints for learning $C(i, :)$. Now if $\{(i,j),(j,i) : j \in [n]\} \cap \Omega = \emptyset$ for some $i \in [n]$, this implies there are no equality constraints to learn $C(i, ;)$. Therefore, there are an infinite number of C matrices such that $A_{ij} = (CC^T)_{ij}, \forall (i,j) \in \Omega$ but $A \neq CC^T$. Further, since $C \in \mathbb{R}^{n \times r}$, it must be **rank**$\{CC^T\} \leq r$, establishing that we can find an infinite number of **rank**$\{r\}$ completions of $A$ from the set of entries indexed in $\Omega$.

The rest of this proof now considers the case when $\mathcal{I}_1 \cup \mathcal{I}_2 = [n]$ but $\mathbf{rank}\{A(\iota, \iota)\} < r$. The matrix $A$ is of rank $r$, and $A(\iota, \iota)$ is a submatrix of $A$. Therefore, it must be that $\mathbf{rank}\{A(\iota, \iota)\} \leq r$. We will show that when $\mathbf{rank}\{A(\iota, \iota)\} < r$ we can find two or more rank $r$ completions for $A$ which match the entries of $A$ in $\Omega$ but differ on their entries outside of $\Omega$, establishing that there is no unique rank $r$ completion for $A$ from the set of observed entries in $\Omega$.

The matrix $A$ is PSD, and we by lemma 1 we can decompose $A$ as $A = CC^T$ for some $C \in \mathbb{R}^{n \times r}$.

Let $n_1 = |\mathcal{I}_1|$ and $n_2 = |\mathcal{I}_2|$. Define $\phi_1$ and $\phi_2$ such that $C_1(\phi_1, :) = C(\iota, :)$ and $C_2(\phi_2, :) = C(\iota, :)$. Define $\eta_2 = [n_2] \setminus \phi_2$. We construct a matrix, $\hat{C} \in \mathbb{R}^{n \times r}$, as

$$\hat{C}(\mathcal{I}_1, :) = C(\mathcal{I}_1, :),$$
$$\hat{C}(\eta_2, :) = C(\eta_2, :)W,$$

for some orthogonal $W$ such that $C(\iota, :) = C(\iota, :)W$.

If $\iota = \emptyset$, then any orthogonal matrix will satisfy this constraint. If $\iota \neq \emptyset$ and rank $\mathbf{rank}\{A(\iota, \iota)\} < r$, then such an orthogonal $W$ still exists and is non-unique. To see this note that if the rank of $A(\iota, \iota)$ is less than $r$, the rank of $C(\iota, :)$ must also be less than $r$ as $C(\iota, :)C(\iota, :)^T = A(\iota, \iota)$. By lemma 2 we can then conclude that there exists a non-unique orthogonal $W$ such that $C(\iota, :) = C(\iota, :)W$.

Let $\hat{A} = \hat{C}\hat{C}^T$. Any such $\hat{A}$ must be of rank $r$ as $\mathbf{rank}\{\hat{C}\} \leq r$. Further, $\hat{A}$ must agree with $A$ on all entries indexed by $\Omega$. To see this note

$$\begin{aligned}
\hat{A}(\mathcal{I}_1, \mathcal{I}_1) &= \hat{C}(\mathcal{I}_1, :)\hat{C}(\mathcal{I}_1, :)^T \\
&= C(\mathcal{I}_1, :)C(\mathcal{I}_1, :)^T \\
&= A(\mathcal{I}_1, \mathcal{I}_1),
\end{aligned}$$

and

$$\begin{aligned}
\hat{A}(\mathcal{I}_2, \mathcal{I}_2) &= \hat{C}(\mathcal{I}_2, :)\hat{C}(\mathcal{I}_2, :)^T \\
&= C(\mathcal{I}_2, :)WW^T C(\mathcal{I}_2, :)^T \\
&= C(\mathcal{I}_2, :)C(\mathcal{I}_2, :)^T \\
&= A(\mathcal{I}_2, \mathcal{I}_2),
\end{aligned}$$

where we have used the relations $\hat{C}(\mathcal{I}_2 \cap \iota, :) = C(\iota, :) = C(\iota, :)W$ and $\hat{C}(\mathcal{I}_2 \setminus \iota, :) = \hat{C}(\eta_2, :) = C(\eta_2, :)W$ to conclude $\hat{C}(\mathcal{I}_2, :) = C(\mathcal{I}_2, :)W$.

However, if $W$ is non-unique the product

$$\hat{A}(\mathcal{I}_1, \mathcal{I}_2) = C(\mathcal{I}_1, :)W^T C(\mathcal{I}_2, :)^T$$

will in general be non-unique, which will yield non-unique completions of the entries of $A$ not indexed by $\Omega$. Further, by assumption $\Omega \neq [n] \times [n]$, so there are one or more entries of $A$ not indexed by $\Omega$. This establishes the existence of a non-unique rank $r$ completion for $A$ from the set of entries indexed in $\Omega$ and completes the proof.

$\square$

**Corollary 7.** *Let $\Omega \neq [n] \times [n]$ index the set of observed entries of a rank $r$ matrix $A \in \mathcal{S}_+^n$. Suppose there exists a $\rho_1, \ldots, \rho_k$ such that $\cup_{i=1}^k \rho_i \times \rho_i = \Omega$ for all $i \in [k]$, $\rho_i \nsubseteq \rho_j$ for all pairs $(i, j) \in [k] \times [k]$, and $\rho_i \cap \rho_j = \emptyset$ for any pair $(i, j) \in \{(i, j) | |i - j| > 1, i \in [k], j \in [k]\}$.*

*Define $\iota_i = \rho_i \cap \rho_{i-1}$ for $i \in \{2, \ldots, k\}$. Then it is necessary that $\cup_{i=1}^k \rho_i = [n]$ and $\mathbf{rank}\{A(\iota_i, \iota_i)\} = r$ for all $i \in \{2, \ldots, k\}$ for there to be a unique rank $r$ completion of $A$.*

*Proof.* Fix $i^* \geq 2$. Define $\rho^*_{i-1} = \cup^{i^*-1}_{j=1}\rho_j$ and $\rho^*_i = \cup^k_{j=i^*}\rho_k$ and $\Omega^* = \rho^*_{i-1} \times \rho^*_{i-1} \cup \rho^*_i \times \rho^*_i$.

By construction $\rho^*_{i-1} \in [n]$ and $\rho^*_i \in [n]$. Further, we prove that $\Omega^* \neq [n] \times [n]$ by contradiction. Assume that $\Omega^* = [n] \times [n]$. Then this would require that either $\rho^*_{i-1} = [n]$ or $\rho^*_i = [n]$, which would imply $\rho^*_{i-1} \cap \rho^*_i = \rho^*_i$ or $\rho^*_{i-1} \cap \rho^*_i = \rho^*_{i-1}$. However, since we assume $\rho_i \cap \rho_j = \emptyset$ for any pair $(i,j) \in \{(i,j)||i-j| > 1, i \in [k], j \in [k]\}$, it must be that $\rho^*_{i-1} \cap \rho^*_i = \rho_{i^*-1} \cap \rho_{i^*}$. However, if $\rho^*_{i-1} \cap \rho^*_i = \rho^*_i$ or $\rho^*_{i-1} \cap \rho^*_i = \rho^*_{i-1}$ and $\rho^*_{i-1} \cap \rho^*_i = \rho_{i^*-1} \cap \rho_{i^*}$, then it must be that $\rho_{i^*} \subseteq \rho_{i^*-1}$ or $\rho_{i^*-1} \subseteq \rho_{i^*}$, which is a contradiction of the assumption that $\rho_i \not\subseteq \rho_j$ for any pair $(i,j) \in [k] \times [k]$.

Thus, we have two sets, $\rho^*_{i-1}$ and $\rho^*_i$, which index $A$ as required by lemma 6. Now assume that $\cup^k_{i=1}\rho_k \neq [n]$. Then $\rho^*_{i-1} \cup \rho^*_i \neq [n]$, so by lemma 6, $A$ cannot be completed from the entries indexed by $\Omega^*$.

Now, assume that $\cup^k_{i=1}\rho_k = [n]$ so $\rho^*_{i-1} \cup \rho^*_i = [n]$, but there exists an $i^*$ such that $\mathbf{rank}\{A(\iota_{i^*}, \iota_{i^*})\} < r$. Define $\iota^* = \rho^*_{i-1} \cap \rho^*_i$.

Then

$$\begin{aligned}
\mathbf{rank}\{A(\iota^*, \iota^*)\} &= \mathbf{rank}\{A(\rho^*_{i-1} \cap \rho^*_i, \rho^*_{i-1} \cap \rho^*_i)\} \\
&= \mathbf{rank}\{A(\rho_{i^*-1} \cap \rho_{i^*}, \rho_{i^*-1} \cap \rho_{i^*})\} \\
&= \mathbf{rank}\{A(\iota_{i^*}, \iota_{i^*})\} \\
&< r.
\end{aligned}$$

Therefore, by lemma 6 there is again no unique rank $r$ completion of $A$ from the entries indexed by $\Omega^*$.

However,

$$\begin{aligned}
\Omega &= \cup^k_{j=1}\rho_j \times \rho_j \\
&= \left(\cup^{i^*-1}_{j=1}\rho_j \times \rho_j\right) \cup \left(\cup^n_{j=i^*}\rho_j \times \rho_j\right) \\
&\subseteq \rho^*_{i-1} \times \rho^*_{i-1} \cup \rho^*_i \times \rho^*_i \\
&= \Omega^*.
\end{aligned}$$

Therefore, the set $\Omega^*$ is a superset of $\Omega$. If no unique rank $r$ completion can be found from a superset of $\Omega$, then the entries indexed by $\Omega$ must themselves contain insufficient information to allow for a unique rank $r$ completion of $A$, completing the proof.

$\square$

**Corollary 8.** *Let $\Omega \neq [n] \times [n]$ index the set of observed entries of a rank $r$ matrix $A \in \mathcal{S}^n_+$. Suppose there exists a $\rho_1, \ldots, \rho_k$ such that $\cup^k_{i=1}\rho_i \times \rho_i = \Omega$, $\rho_i \not\subseteq \rho_j$ for all pairs $(i,j) \in [k] \times [k]$ and $\rho_i \cap \rho_j = \emptyset$ for any pair $(i,j) \in \{(i,j)|i \neq 1, j \neq 1, i \in [k], j \in [k]\}$.*

*Define $\iota_i = \rho_i \cap \rho_1$ for $i \in \{2, \ldots, k\}$. Then it is necessary that $\cup^k_{i=1}\rho_i = [n]$ and $\mathbf{rank}\{A(\iota_i, \iota_i)\} = r$ for all $i \in \{2, \ldots, k\}$ for there to be a unique rank $r$ completion of $A$.*

*Proof.* Fix $i^* \geq 2$. Define $\rho^*_{i-1} = \cup^{i^*-1}_{j=1}\rho_j$ and $\rho^*_i = \cup^k_{j=i^*}\rho_k$ and $\Omega^* = \rho^*_{i-1} \times \rho^*_{i-1} \cup \rho^*_i \times \rho^*_i$.

By construction $\rho^*_{i-1} \in [n]$ and $\rho^*_i \in [n]$. Further, we prove that $\Omega^* \neq [n] \times [n]$ by contradiction. Assume that $\Omega^* = [n] \times [n]$. Then this would require that either $\rho^*_{i-1} = [n]$ or $\rho^*_i = [n]$, which would imply $\rho^*_{i-1} \cap \rho^*_i = \rho^*_i$ or $\rho^*_{i-1} \cap \rho^*_i = \rho^*_{i-1}$. However, since we assume $\rho_i \cap \rho_j = \emptyset$ for any pair $(i,j) \in \{(i,j)|i \neq 1, j \neq 1, i \in [k], j \in [k]\}$, this would imply that $\rho_1 \cap \rho_i = \rho_i$ for some $i \geq 2$ or $\rho_1 \cap \rho_i = \rho_1$ for some $i \geq 2$, implying that $\rho_i \subseteq \rho_1$ for some $i \geq 2$ or $\rho_1 \subseteq \rho_i$ for some $i \geq 2$, both of which contradict the assumption that $\rho_i \not\subseteq \rho_j$ for any pair $(i,j) \in [k] \times [k]$.

Thus, we have two sets, $\rho_{i-1}^*$ and $\rho_i^*$, which index $A$ as required by lemma 6. Now assume that $\cup_{i=1}^k \rho_k \neq [n]$. Then $\rho_{i-1}^* \cup \rho_i^* \neq [n]$, so by lemma 6, $A$ cannot be completed from the entries indexed by $\Omega^*$.

Now, assume that $\cup_{i=1}^k \rho_k = [n]$ so $\rho_{i-1}^* \cup \rho_i^* = [n]$, but there exists an $i^*$ such that $\mathbf{rank}\{A(\iota_{i^*}, \iota_{i^*})\} < r$. Define $\iota^* = \rho_{i-1}^* \cap \rho_i^*$.

Then

$$
\begin{aligned}
\mathbf{rank}\{A(\iota^*, \iota^*)\} &= \mathbf{rank}\{A(\rho_{i-1}^* \cap \rho_i^*, \rho_{i-1}^* \cap \rho_i^*)\} \\
&= \mathbf{rank}\{A(\rho_1 \cap \rho_{i^*}, \rho_1 \cap \rho_{i^*})\} \\
&= \mathbf{rank}\{A(\iota_{i^*}, \iota_{i^*})\} \\
&< r.
\end{aligned}
$$

Therefore, by lemma 6 there is again no unique rank $r$ completion of $A$ from the entries indexed by $\Omega^*$. However,

$$
\begin{aligned}
\Omega &= \cup_{j=1}^k \rho_j \times \rho_j \\
&= \left(\cup_{j=1}^{i^*-1} \rho_j \times \rho_j\right) \cup \left(\cup_{j=i^*}^n \rho_j \times \rho_j\right) \\
&\subseteq \rho_{i-1}^* \times \rho_{i-1}^* \cup \rho_i^* \times \rho_i^* \\
&= \Omega^*.
\end{aligned}
$$

Therefore, the set $\Omega^*$ is a superset of $\Omega$. If no unique rank $r$ completion can be found from a superset of $\Omega$, then the entries indexed by $\Omega$ must themselves contain insufficient information to allow for a unique rank $r$ completion of $A$, completing the proof. $\qquad\square$

**Corollary 9.** *Let $\Omega \neq [n] \times [n]$ index the set of observed entries of a rank $r$ matrix $A \in \mathcal{S}_+^n$. Assume there exists an $\mathcal{I} \subseteq [n]$ and $\rho \subseteq [n]$ such that $\rho \times \rho \subseteq \Omega$, $\rho \nsubseteq \mathcal{I}$ and $\Omega \subseteq \rho \times \rho \cup \mathcal{I} \times \mathcal{I}$.*

*Define $\iota = \rho \cap \mathcal{I}$. Then it is necessary that $\rho \cup \mathcal{I} = [n]$ and $\mathbf{rank}\{A(\iota, \iota)\} = r$ for all $i \in \{2, \ldots, k\}$ for there to be a unique rank $r$ completion of $A$.*

*Proof.* The proof is a straightforward application of lemma 6.

Define $\Omega^* = \rho \times \rho \cup \mathcal{I}_1 \times \mathcal{I}_1$. Assume $\rho \cup \mathcal{I}_1 \neq [n]$. Then by lemma 6, $A$ cannot be recovered from the entries indexed by $\Omega^*$.

Now assume that $\rho \cup \mathcal{I}_1 = [n]$, but $\mathbf{rank}\{A(\iota, \iota)\} < r$. Then by lemma 6, $A$ cannot be recovered from the entries indexed by $\Omega^*$.

However, $\Omega^* \supseteq \Omega$. If no unique rank $r$ completion can be found from a superset of $\Omega$, then the entries indexed by $\Omega$ must themselves contain insufficient information to allow for a unique rank $r$ completion of $A$, completing the proof.

$\qquad\square$

**Lemma 10.** *For any matrix pair $C \in \mathbb{R}^{n \times p}$ and $\hat{C} \in \mathbb{R}^{n \times p}$ such that $\min_{W:WW^T=I} \left\| C - \hat{C}W \right\|_F \leq \epsilon$, there exists an $P \in \mathbb{R}^{n \times p}$ and orthogonal $U \in \mathbb{R}^{p \times p}$ such that $\hat{C} = CU + P$ and $\|P\|_F \leq \epsilon$.*

*Proof.* Define $U^T = \operatorname{argmin}_{W:WW^T=I} \left\| C - \hat{C}W \right\|_F$. Then by assumption $\left\| C - \hat{C}U^T \right\|_F \leq \epsilon$. Further, the set of orthogonal matrices of size 1 or greater is nonempty so such a $U$ must always exist. Now define $\hat{C} = CU + P$. Because we can pick any $P \in \mathbb{R}^{n \times r}$, such a $P$ always exists. Then

$$\|P\|_F = \|CU - CU - P\|_F$$
$$= \left\|CU - \hat{C}\right\|_F$$
$$= \left\|C - \hat{C}U^T\right\|_F$$
$$\leq \epsilon.$$

$\square$

**Lemma 11.** *For $C$ and $\hat{C}$ in $\mathbb{R}^{n \times p}$, let $\min_{W:WW^T=I} \left\|C - \hat{C}W\right\|_F \leq \epsilon$ and $\|C\|_F \leq \delta$. Then*

$$\left\|CC^T - \hat{C}\hat{C}^T\right\|_F \leq 2\delta\epsilon + \epsilon^2.$$

*Proof.* For any $W : WW^T = I$, we can write

$$\left\|\hat{C}\right\|_F = \left\|\hat{C}W\right\|_F = \left\|\hat{C}W - C + C\right\|_F$$
$$\leq \left\|\hat{C}W - C\right\|_F + \|C\|_F$$
$$= \left\|C - \hat{C}W\right\|_F + \|C\|_F,$$

where we have made use of triangle inequality for norms and the unitarily invariance of the Frobenius norm. The above holds for any $W$. Selecting $W^* = \operatorname{argmin}_{W:WW^T=I} \left\|C - \hat{C}W\right\|_F$, we have

$$\left\|\hat{C}\right\|_F \leq \left\|C - \hat{C}W^*\right\|_F + \|C\|_F$$
$$\leq \epsilon + \delta.$$

Again, for any $W : WW^T = I$, we can write

$$\left\|CC^T - \hat{C}\hat{C}^T\right\|_F = \left\|CC^T - \hat{C}WW^T\hat{C}^T\right\|_F$$
$$= \left\|CC^T - \hat{C}WC^T + \hat{C}WC^T - \hat{C}WW^T\hat{C}^T\right\|_F$$
$$= \left\|[C - \hat{C}W]C^T + \hat{C}W[C^T - W^T\hat{C}^T]\right\|_F$$
$$\leq \left\|[C - \hat{C}W]C^T\right\|_F + \left\|\hat{C}W[C^T - W^T\hat{C}^T]\right\|_F$$
$$\leq \|C\|_F \left\|C - \hat{C}W\right\|_F + \left\|\hat{C}W\right\|_F \left\|C - \hat{C}W\right\|_F$$
$$= \|C\|_F \left\|C - \hat{C}W\right\|_F + \left\|\hat{C}\right\|_F \left\|C - \hat{C}W\right\|_F,$$

where in addition to the two properties of the Frobenius norm we used before we have also used the submultiplicative property of this norm. We can again pick $W$ arbitrarily, and we again pick $W^* = \operatorname{argmin}_{W:WW^T=I} \left\|C - \hat{C}W\right\|_F$, so that

$$\left\| CC^T - \hat{C}\hat{C}^T \right\|_F \leq \|C\|_F \left\| C - \hat{C}W^* \right\|_F + \left\| \hat{C} \right\|_F \left\| C - \hat{C}W^* \right\|_F$$
$$\leq \|C\|_F \, \epsilon + \|\hat{C}\|_F \epsilon$$
$$\leq \delta\epsilon + (\epsilon + \delta)\epsilon$$
$$= 2\delta\epsilon + \epsilon^2,$$

where we have used the bound on $\|\hat{C}\|_F$ in moving to the second to last line of the proof. $\quad\square$

**Lemma 12.** *For $C_1, C_2 \in \mathbb{R}^{n \times p}$, such that $\|C_1\|_F \leq \delta_1$ and $\|C_2\|_F \leq \delta_2$ define*

$$\hat{C}_1 = C_1 + P_1$$
$$\hat{C}_2 = C_2 + P_2,$$

*for some error matrices $P_1, P_2 \in \mathbb{R}^{n \times p}$ such that $\|P_1\|_F \leq \epsilon_1$ and $\|P_2\|_F \leq \epsilon_2$. Then*

$$\left\| C_1^T C_2 - \hat{C}_1^T \hat{C}_2 \right\|_F \leq \epsilon_1 \delta_2 + \epsilon_1 \epsilon_2 + \delta_1 \epsilon_2.$$

*Proof.* Throughout this proof we will use the triangle inequality of norms and the submultiplicative property of the Frobenius norm without further comment.

First, note that

$$\left\| \hat{C}_1 \right\|_F = \left\| \hat{C}_1 - C_1 + C_1 \right\|_F$$
$$\leq \left\| \hat{C}_1 - C_1 \right\|_F + \|C_1\|_F$$
$$= \|P_1\|_F + \|C_1\|_F$$
$$\leq \epsilon_1 + \delta_1.$$

Next, we have

$$\left\| C_1^T C_2 - \hat{C}_1^T \hat{C}_2 \right\|_F = \left\| C_1^T C_2 - \hat{C}_1^T C_2 + \hat{C}_1^T C_2 - \hat{C}_1^T \hat{C}_2 \right\|_F$$
$$= \left\| \left[ C_1^T - \hat{C}_1^T \right] C_2 + \hat{C}_1^T \left[ C_2 - \hat{C}_2 \right] \right\|_F$$
$$= \left\| -P_1^T C_2 - \hat{C}_1^T P_2 \right\|_F$$
$$\leq \left\| P_1^T C_2 \right\|_F + \left\| \hat{C}_1^T P_2 \right\|_F$$
$$\leq \|P_1\|_F \|C_2\|_F + \|\hat{C}_1\|_F \|P_2\|_F$$
$$\leq \epsilon_1 \delta_2 + [\epsilon_1 + \delta_1]\epsilon_2$$
$$= \epsilon_1 \delta_2 + \epsilon_1 \epsilon_2 + \delta_1 \epsilon_2,$$

where he have used the bound on $\left\| \hat{C}_1 \right\|$ in moving to the second to last line of the proof. $\quad\square$

**Lemma 13.** *Let $U$ be an $n \times n$ orthogonal matrix. For the unit length vector $v \in \mathbb{R}^n$, define $\theta_i(v) = \min_{k:|k|=1} \cos^{-1}(kU(:,i)^T v)$, where $|\cdot|$ indicates absolute value. Then*

$$\sum_{j \neq i} (U(:,j)^T v)^2 = \sin^2 \theta_i(v).$$

*Proof.* The matrix $U$ is orthogonal, so it must be that

$$\sum_j (U(:,j)^T v)^2 = v^T U U^T v = v^T v = \|v\|_2^2 = 1,$$

where $\|\cdot\|_2$ indicates the $\ell_2$-norm. Therefore, for $k \in \{1, -1\}$, we have

$$\sum_j (U(:,j)^T v)^2 = (kU(:,i)^T v)^2 + \sum_{j \neq i} (U(:,j)^T v)^2 = \cos^2 \theta_i(v) + \sum_{j \neq i} (U(:,j)^T v)^2 = 1,$$

which we can use to find

$$\sum_{j \neq i} (U(:,j)^T v)^2 = 1 - \cos^2 \theta_i(v) = \sin^2 \theta_i(v).$$

$\square$

**Lemma 14.** *Consider the matrices $A, \tilde{A} \in \mathcal{S}_+^n$ of rank $r$ and $\tilde{r} \geq r$, respectively. Define $\hat{C} \in \mathbb{R}^{n \times r}$ column-wise such that*

$$\hat{C}(:,i) = \tilde{\lambda}_i^{1/2} \tilde{E}(:,i), \qquad \forall i \leq r. \tag{5}$$

*For $i \leq r$, define $\theta_i = \min_{k_i : |k_i| = 1} \cos^{-1}(k_i \tilde{E}_i^T E_i)$, where $|\cdot|$ indicates absolute value.*

*Then for any $C \in \mathbb{R}^{n \times r}$ such that $A = CC^T$ and $\tilde{A}$*

$$\min_{W:WW^T=I} \|C - \hat{C}W\|_F \leq \sqrt{\sum_{i=1}^r |\lambda_i - \tilde{\lambda}_i| + 4\lambda_i \sin^2(\theta_i/2) + 4\lambda_i^{1/2} |\lambda_i - \tilde{\lambda}_i|^{1/2} \sin^2(\theta_i/2)}.$$

*Proof.* Let $\mathcal{S}$ be the set of $r \times r$ diagonal sign matrices (diagonal matrices with values of $1$ or $-1$ along the diagonal). The Frobenius norm is unitarily invariant, so for any $C$ and $\hat{C}$,

$$\min_{W:WW^T=I} \|C - \hat{C}W\|_F = \min_{W:WW^T=I} \|CW - \hat{C}\|_F$$
$$= \min_{W:WW^T=I, S \in \mathcal{S}} \|CWS - \hat{C}S\|_F.$$

Further, for orthogonal $W$, the matrix $WS^{-1}$ is also orthogonal, so if $W^*$ minimizes $\min_{W:WW^T=I, S \in \mathcal{S}} \|CW - \hat{C}S\|_F$, then $W^* S^{-1}$ is an orthogonal matrix that minimizes $\min_{W:WW^T=I, S \in \mathcal{S}} \|CWS - \hat{C}S\|_F$, establishing that

$$\min_{W:WW^T=I} \|C - \hat{C}W\|_F = \min_{W:WW^T=I, S \in \mathcal{S}} \|CW - \hat{C}S\|_F.$$

Now select $W'$ such that $CW' = E\Lambda^{1/2}$. If $CC^T = A$, such a $W'$ is guaranteed to always exist by lemma 4. Next, note that

$$\min_{W:WW^T=I,S\in\mathcal{S}}\|CW-\hat{C}S\|_F \leq \min_{S\in\mathcal{S}}\|CW'-\hat{C}S\|_F$$

$$= \min_{S\in\mathcal{S}}\|E\Lambda^{1/2}-\hat{C}S\|_F.$$

We next can write

$$\min_{S\in\mathcal{S}}\|E\Lambda^{1/2}-\hat{C}S\|_F^2 = \min_{S\in\mathcal{S}}\sum_{i=1}^{r}\|E(:,i)\lambda_i^{1/2}-\hat{C}(:,i)S(i,i)\|_2^2$$

$$= \sum_{i=1}^{r}\min_{k_i:k_i\in\{-1,1\}}\|E(:,i)\lambda_i^{1/2}-k_i\tilde{E}(:,i)\tilde{\lambda}_i^{1/2}\|_2^2, \qquad (6)$$

where in moving the last line above we have used (5) to substitute for $\hat{C}(:,i)$ and written the minimization over $S\in\mathcal{S}$ equivalently in terms of individual $k_i\in\{-1,1\}$.

Now, let $\tilde{E}_\perp$ be a matrix with orthonormal columns that span the subspace of $\mathbb{R}^n$ orthogonal to the subspace spanned by the first $r$ columns of $\tilde{E}$. The columns of the unitary matrix $B = [\tilde{E}_1,\ldots,\tilde{E}_k,\tilde{E}_\perp]$ then form an orthonormal basis for $\mathbb{R}^n$. For any vector $x\in\mathbb{R}^n$, it must then be that $\|x\|_2 = \|B^Tx\|_2$. Therefore

$$\sum_{i=1}^{r}\min_{k_i:k_i\in\{-1,1\}}\|E(:,i)\lambda_i^{1/2}-k_i\tilde{E}(:,i)\tilde{\lambda}_i^{1/2}\|_2^2 = \sum_{i=1}^{r}\min_{k_i:k_i\in\{-1,1\}}\|B^T[E(:,i)\lambda_i^{1/2}-k_i\tilde{E}(:,i)\tilde{\lambda}_i^{1/2}]\|_2^2$$

$$= \sum_{i=1}^{r}\min_{k_i:k_i\in\{-1,1\}}\sum_{j=1}^{n}\left(B(:,j)^T[E(:,i)\lambda_i^{1/2}-k_i\tilde{E}(:,i)\tilde{\lambda}_i^{1/2}]\right)^2$$

$$= \sum_{i=1}^{r}\min_{k_i:k_i\in\{-1,1\}}\left(\left(B(:,i)^T[E(:,i)\lambda_i^{1/2}-k_i\tilde{E}(:,i)\tilde{\lambda}_i^{1/2}]\right)^2+\ldots\right.$$

$$\left.\sum_{j\neq i}\left(B(:,j)^T[E(:,i)\lambda_i^{1/2}-k_i\tilde{E}(:,i)\tilde{\lambda}_i^{1/2}]\right)^2\right)$$

$$= \sum_{i=1}^{r}\min_{k_i:k_i\in\{-1,1\}}\left(\left(\lambda_i^{1/2}\tilde{E}(:,i)^TE(:,i)-k_i\tilde{\lambda}_i^{1/2}\right)^2+\ldots\right.$$

$$\left.\sum_{j\neq i}\left(\lambda_i^{1/2}B(:,j)^TE(:,i)\right)^2\right),$$

where in moving to the last equality statement we have recognized that the first $r$ columns of $B$ are by construction the first $r$ columns of $\tilde{E}$.

Using lemma 13, it must be that

$$\sum_{j\neq i}\left(\lambda_i^{1/2}B(:,j)^TE(:,i)\right)^2 = \lambda_i\sum_{j\neq i}\left(B(:,j)^TE(:,i)\right)^2$$

$$= \lambda_i\sin^2\theta_i,$$

where $\theta_i = \min_{k_i:|k_i|=1}\cos^{-1}(k_iB(:,i)^TE(:,i)) = \min_{k_i:|k_i|=1}\cos^{-1}(k_i\tilde{E}(:,i)^TE(:,i))$.

Further, note that

$$\min_{k_i : k_i \in \{-1,1\}} \left( \lambda_i^{1/2} \tilde{E}(:,i)^T E(:,i) - k_i \tilde{\lambda}_i^{1/2} \right)^2 = \min_{k_i : k_i \in \{-1,1\}} \left( k_i \lambda_i^{1/2} \tilde{E}(:,i)^T E(:,i) - \tilde{\lambda}_i^{1/2} \right)^2.$$

Additionally, $\lambda_i^{1/2} \geq 0$ and $\tilde{\lambda}_i^{1/2} \geq 0$ since both $A$ and $\tilde{A}$ are PSD. Therefore, the $k_i$ that minimizes the above equation must render $k_i \tilde{E}(:,i)^T E(:,i) \geq 0$, from which we can conclude that $k_i \tilde{E}(:,i)^T E(:,i) = \cos(\theta_i)$, when $\theta_i = \min_{k_i : |k_i|=1} \cos^{-1}(k_i \tilde{E}(:,i)^T E(:,i))$, so

$$\min_{k_i : k_i \in \{-1,1\}} \left( k_i \lambda_i^{1/2} \tilde{E}(:,i)^T E(:,i) - \tilde{\lambda}_i^{1/2} \right)^2 = \left( \lambda_i^{1/2} \cos(\theta_i) - \tilde{\lambda}_i^{1/2} \right)^2.$$

We can therefore write (6) as

$$\sum_{i=1}^{r} \min_{k_i : k_i \in \{-1,1\}} \| E(:,i) \lambda_i^{1/2} - k_i \tilde{E}(:,i) \tilde{\lambda}_i^{1/2} \|_2^2 = \sum_{i=1}^{r} \left[ \left( \lambda_i^{1/2} \cos(\theta_i) - \tilde{\lambda}_i^{1/2} \right)^2 + \lambda_i \sin^2 \theta_i \right]$$

$$= \sum_{i=1}^{r} \left[ \lambda_i + \tilde{\lambda}_i - 2 \cos(\theta_i) \lambda_i^{1/2} \tilde{\lambda}_i^{1/2} \right].$$

We now bound

$$\lambda_i + \tilde{\lambda}_i - 2 \cos(\theta_i) \lambda_i^{1/2} \tilde{\lambda}_i^{1/2} = (\lambda_i^{1/2} - \tilde{\lambda}_i^{1/2})^2 + 2 \lambda_i^{1/2} \tilde{\lambda}_i^{1/2} - 2 \cos(\theta_i) \lambda_i^{1/2} \tilde{\lambda}_i^{1/2}$$

$$= (\lambda_i^{1/2} - \tilde{\lambda}_i^{1/2})^2 + 2 \lambda_i^{1/2} \tilde{\lambda}_i^{1/2} \left( 1 - \cos(\theta_i) \right).$$

Note that

$$(\lambda_i^{1/2} - \tilde{\lambda}_i^{1/2})^2 = \frac{(\lambda_i - \tilde{\lambda}_i)^2}{(\lambda_i^{1/2} + \tilde{\lambda}_i^{1/2})^2}$$

$$= |\lambda_i - \tilde{\lambda}_i| \frac{|\lambda_i - \tilde{\lambda}_i|}{(\lambda_i + \tilde{\lambda}_i + 2 \lambda_i^{1/2} \tilde{\lambda}_i^{1/2})}$$

$$\leq |\lambda_i - \tilde{\lambda}_i| \frac{|\lambda_i - \tilde{\lambda}_i|}{\lambda_i + \tilde{\lambda}_i}$$

$$\leq |\lambda_i - \tilde{\lambda}_i|,$$

and we can use a half angle formula to write $\cos(\theta_i) = 1 - 2 \sin^2(\theta_i/2)$, and we have

$$\lambda_i + \tilde{\lambda}_i - 2 \cos(\theta_i) \lambda_i^{1/2} \tilde{\lambda}_i^{1/2} \leq |\lambda_i - \tilde{\lambda}_i| + 4 \lambda_i^{1/2} \tilde{\lambda}_i^{1/2} \sin^2(\theta_i/2).$$

Further note that $\tilde{\lambda}_i^{1/2} = \lambda_i^{1/2} + \tilde{\lambda}_i^{1/2} - \lambda_i^{1/2} \leq \lambda_i^{1/2} + |\lambda_i^{1/2} - \tilde{\lambda}_i^{1/2}|$, so we have

$$\lambda_i + \tilde{\lambda}_i - 2 \cos(\theta_i) \lambda_i^{1/2} \tilde{\lambda}_i^{1/2} \leq |\lambda_i - \tilde{\lambda}_i| + 4 \lambda_i^{1/2} (\lambda_i^{1/2} + |\lambda_i^{1/2} - \tilde{\lambda}_i^{1/2}|) \sin^2(\theta_i/2)$$

$$= |\lambda_i - \tilde{\lambda}_i| + 4 \lambda_i \sin^2(\theta_i/2) + 4 \lambda_i^{1/2} |\lambda_i^{1/2} - \tilde{\lambda}_i^{1/2}| \sin^2(\theta_i/2)$$

$$\leq |\lambda_i - \tilde{\lambda}_i| + 4 \lambda_i \sin^2(\theta_i/2) + 4 \lambda_i^{1/2} |\lambda_i - \tilde{\lambda}_i|^{1/2} \sin^2(\theta_i/2),$$

where in moving to the last line we have again used the inequality $(\lambda_i^{1/2} - \tilde{\lambda}_i^{1/2})^2 \leq |\lambda_i - \tilde{\lambda}_i|$. We can then complete the proof by writing

$$\min_{W:WW^T=I} \|C - \hat{C}W\|_F^2 \leq \sum_{i=1}^{r} |\lambda_i - \tilde{\lambda}_i| + 4\lambda_i \sin^2(\theta_i/2) + 4\lambda_i^{1/2}|\lambda_i - \tilde{\lambda}_i|^{1/2} \sin^2(\theta_i/2),$$

where the final result is obtained by taking the square root of both sides of the above inequality.

$\square$

**Lemma 15.** *Consider the matrices $A, \tilde{A} \in \mathcal{S}_+^n$ of rank $r$ and $\tilde{r} \geq r$, respectively. Define $S = \tilde{A} - A$. Let $\delta = \min\{\min_{i \in [r-1]}, |\lambda_i - \lambda_{i+1}|, \lambda_r\}$. Let $\|S\|_F \leq \epsilon$, for some $\epsilon < \delta/2 \leq 1$. Define $\hat{C} \in \mathbb{R}^{n \times r}$ column-wise as*

$$\hat{C}(:,i) = \tilde{\lambda}_i^{1/2} \tilde{E}(:,i), \qquad \forall i \leq r. \tag{7}$$

*Then for any $C$ such that $CC^T = A$,*

$$\min_{W:WW^T=I} \|C - \hat{C}W\|_F \leq K\sqrt{r}\epsilon,$$

*for $K = \sqrt{1 + \frac{16\lambda_1}{\delta^2} + \frac{8\sqrt{2}\lambda_1^{1/2}}{\delta^{3/2}}}$.*

*Proof.* For $i \leq r$, define $\theta_i = \min_{k_i:|k_i|=1} \cos^{-1}(k\tilde{E}_i^T E_i)$. We start by using lemma 14 to write

$$\min_{W:WW^T=I} \|C - \hat{C}W\|_F \leq \sqrt{\sum_{i=1}^{r} |\lambda_i - \tilde{\lambda}_i| + 4\lambda_i \sin^2(\theta_i/2) + 4\lambda_i^{1/2}|\lambda_i - \tilde{\lambda}_i|^{1/2} \sin^2(\theta_i/2)}. \tag{8}$$

We will now proceed to establish our theorem by using results in matrix perturbation theory to bound the different terms in the square root of equation (8). Throughout this proof it should be understood that $\lambda_{r+1}, \ldots, \lambda_n = 0$ and $\tilde{\lambda}_{\tilde{r}+1}, \ldots, \tilde{\lambda}_n = 0$).

By construction $S$ is a real, symmetric matrix, and we begin by using Weyl's theorem, which states that $\tilde{\lambda}_i \in [\lambda_i + \lambda_1(S), \lambda_i + \lambda_n(S)]$ [1], where $\lambda_i(S)$ indicates the $i^{th}$ eigenvalue of $S$, to bound $|\lambda_i - \tilde{\lambda}_i| \leq \max_{j \in \{1,n\}} |\lambda_j(S)| \leq \max_{j \in [n]} |\lambda_j(S)|$. We then have

$$\sum_{i=1}^{r} |\lambda_i - \tilde{\lambda}_i| \leq \sum_{i=1}^{r} \max_{j \in [n]} |\lambda_j(S)| = r \max_{j \in [n]} |\lambda_j(S)| = r\|S\|_2 \leq r \|S\|_F \leq r\epsilon, \tag{9}$$

where $\|\cdot\|_2$ indicates the spectral norm.

We next bound $\sin^2(\theta_i/2)$ for $i \leq r$. For $\tilde{E}(:,i)$, we define

$$r_i = A\tilde{E}(:,i) - \tilde{\lambda}_i \tilde{E}(:,i), \tag{10}$$

which is a residual vector that will be 0 if and only if $(\tilde{\lambda}_i, \tilde{E}(:,i))$ are an eigenpair of $A$.

Now define

$$\mathbf{sep}\{A, \tilde{A}\} = \min_{i \in [r]} \min_{j \in [n], j \neq i} |\tilde{\lambda}_i - \lambda_j|. \tag{11}$$

By the Sin Theta theorem (c.f., theorem 3.4 in §V of [1]), we know that

$$|\sin(\theta_i)| \leq \frac{\|r_i\|_2}{\min_{j \in [n], j \neq i} |\lambda_j - \tilde{\lambda}_i|}.$$

We also know $\mathbf{sep}\{A, \tilde{A}\} \leq \min_{j \in [n], j \neq i} |\lambda_j - \tilde{\lambda}_i|$, for all $i \in [r]$, so we have

$$\sum_{i=1}^{r} \sin^2(\theta_i) \leq \sum_{i=1}^{r} \left( \frac{\|r_i\|_2}{\mathbf{sep}\{A, \tilde{A}\}} \right)^2 \leq \frac{1}{\mathbf{sep}^2\{A, \tilde{A}\}} \sum_{i=1}^{r} \|r_i\|_2^2.$$

We now bound $\sum_{i=1}^{r} \|r_i\|_2^2$. To simplify things note that

$$r_i = A\tilde{E}(:,i) - \tilde{\lambda}_i \tilde{E}(:,i) = A\tilde{E}(:,i) - \tilde{A}\tilde{E}(:,i) = A\tilde{E}(:,i) - (A+S)\tilde{E}(:,i) = -S\tilde{E}(:,i),$$

so that $\|r_i\|_2 = \left\| S\tilde{E}(:,i) \right\|_2$. We can now write

$$\begin{aligned}
\sum_{i=1}^{r} \|r_i\|_2^2 &= \sum_{i=1}^{r} \left\| S\tilde{E}(:,i) \right\|_2^2 \\
&= \left\| S\tilde{E}(:,1:r) \right\|_F^2 \\
&\leq \left\| S[\tilde{E}(:,1:r), \tilde{E}(:,1:r)_\perp] \right\|_F^2 \\
&= \|S\|_F^2 \\
&\leq \epsilon^2,
\end{aligned}$$

where $\tilde{E}(:,1:r)_\perp$ is a matrix with orthonormal columns spanning a subspace orthogonal to the subspace spanned by the columns of the matrix $\tilde{E}(:,1:r)$, so $[\tilde{E}(:,1:r), \tilde{E}(:,1:r)_\perp]$ is an orthogonal matrix.

Therefore, we have

$$\sum_{i=1}^{r} \sin^2(\theta_i) \leq \frac{\epsilon^2}{\mathbf{sep}^2\{A, \tilde{A}\}}. \tag{12}$$

To complete our bound on $\sum_{i=1}^{r} \sin^2(\theta_i)$, we have only to lower bound $\mathbf{sep}\{A, \tilde{A}\}$.

Let $d_{\max} = \max_{i \in [r]} |\lambda_i - \tilde{\lambda}_i|$. We can again use Weyl's theorem to bound

$$d_{\max} = \max_{i \in [r]} |\lambda_i - \tilde{\lambda}_i| \leq \|S\|_2 \leq \|S\|_F \leq \epsilon < \delta/2. \tag{13}$$

For $i \in [r]$ we also derive an inequality that will be immediately useful. We start by stating,

$$\min_{j \in [n], j \neq i} |\lambda_i - \lambda_j| = \min\left\{ |\lambda_{i-1} - \lambda_i|, |\lambda_{i+1} - \lambda_i| \right\},$$

where is understood the appropriate terms disappear if $i = 1$ or $i = n$. We finish the derivation of the inequality by writing for all $i \in [r]$,

$$\min\left\{|\lambda_{i-1} - \lambda_i|, |\lambda_{i+1} - \lambda_i|\right\} \le \min\{\min_{i \in [r-1],} |\lambda_i - \lambda_{i+1}|, \lambda_r\} = \delta,$$

so we can conclude for all $i \in [r]$,

$$\min_{j \in [n], j \ne i} |\lambda_i - \lambda_j| \le \delta. \tag{14}$$

For all $i \in [r]$, we can then use 13 and 14 to bound

$$
\begin{aligned}
\min_{j \in [n], j \ne i} |\lambda_j - \tilde{\lambda}_i| &= \min_{j \in [n], j \ne i} |\tilde{\lambda}_i - \lambda_j| \\
&= \min_{j \in [n], j \ne i} |\tilde{\lambda}_i - \lambda_i + \lambda_i - \lambda_j| \\
&\ge \min_{j \in [n], j \ne i} |\lambda_i - \lambda_j| - |\tilde{\lambda}_i - \lambda_i| \\
&\ge \min_{j \in [n], j \ne i} \delta - |\tilde{\lambda}_i - \lambda_i| \\
&\ge \min_{j \in [n], j \ne i} \delta - d_{\max} \\
&> \delta - \delta/2 \\
&= \delta/2.
\end{aligned}
$$

Therefore, it must be that $\mathbf{sep}\{A, \tilde{A}\} > \delta/2$. We can then pick up from (12) and complete our bound on $\sum_{i=1}^r \sin^2(\theta_i)$ as

$$\sum_{i=1}^r \sin^2(\theta_i) < \frac{4\epsilon^2}{\delta^2}. \tag{15}$$

We now pick back up with (8) and write

$$
\begin{aligned}
\min_{W:WW^T=I} \|C - \hat{C}W\|_F &\le \sqrt{\sum_{i=1}^r |\lambda_i - \tilde{\lambda}_i| + 4\lambda_i \sin^2(\theta_i/2) + 4\lambda_i^{1/2}|\lambda_i - \tilde{\lambda}_i|^{1/2} \sin^2(\theta_i/2)} \\
&= \sqrt{\sum_{i=1}^r |\lambda_i - \tilde{\lambda}_i| + 4\sum_{i=1}^r \lambda_i \sin^2(\theta_i/2) + 4\sum_{i=1}^r \lambda_i^{1/2}|\lambda_i - \tilde{\lambda}_i|^{1/2} \sin^2(\theta_i/2)} \\
&\le \sqrt{\sum_{i=1}^r |\lambda_i - \tilde{\lambda}_i| + 4\lambda_1 \sum_{i=1}^r \sin^2(\theta_i/2) + 4\lambda_1^{1/2} d_{\max}^{1/2} \sum_{i=1}^r \sin^2(\theta_i/2)}.
\end{aligned}
$$

Further, note that $\theta_i \in [0, \pi/2]$, and therefore $\sin(\theta_i/2) \le \sin(\theta_i)$, so we have

$$\min_{W:WW^T=I} \|C - \hat{C}W\|_F \le \sqrt{\sum_{i=1}^r |\lambda_i - \tilde{\lambda}_i| + 4\lambda_1 \sum_{i=1}^r \sin^2(\theta_i) + 4\lambda_1^{1/2} d_{\max}^{1/2} \sum_{i=1}^r \sin^2(\theta_i)}.$$

We now use (9), (13) and (15) to find

$$\min_{W:WW^T=I} \|C - \hat{C}W\|_F \leq \sqrt{r\epsilon + \frac{16\lambda_1\epsilon^2}{\delta^2} + \frac{8\sqrt{2}\lambda_1^{1/2}\epsilon^2}{\delta^{3/2}}}.$$

For $r \geq 1$ and $\epsilon \leq 1$, we then have

$$\min_{W:WW^T=I} \|C - \hat{C}W\|_F \leq K\sqrt{r\epsilon},$$

for $K = \sqrt{1 + \frac{16\lambda_1}{\delta^2} + \frac{8\sqrt{2}\lambda_1^{1/2}}{\delta^{3/2}}}.$

$\square$

**Lemma 16.** *Let $M$ be a rank $r$ matrix in $\mathbb{R}^{n \times r}$ such that $\|M\|_F \leq \delta$. Define*

$$M_1 = MW_1 + P_1$$
$$M_2 = MW_2 + P_2,$$

*for perturbation matrices $P_1, P_2 \in \mathbb{R}^{n \times r}$ such that $\|P_1\|_F \leq \epsilon_1$ and $\|P_2\|_F \leq \epsilon_2$ and orthonormal matrices $W_1 \in \mathbb{R}^{r \times r}$ and $W_2 \in \mathbb{R}^{r \times r}$.*

*Find*

$$T = \operatorname*{argmin}_{W:WW^T=I} \|MW_1 - MW_2W\|_F^2, \tag{16}$$

*and*

$$\hat{T} = \operatorname*{argmin}_{W:WW^T=I} \|M_1 - M_2W\|_F^2. \tag{17}$$

*Then $T = W_2^T W_1$ is unique. If $M_2^T M_1$ is also of rank $r$, $\hat{T}$ is also unique. Finally, when $M_2^T M_1$ is of rank $r$,*

$$\left\|T - \hat{T}\right\|_F \leq \frac{2}{\sigma_r^2(M)}[\delta(\epsilon_1 + \epsilon_2) + \epsilon_1\epsilon_2].$$

*Proof.* Define $S = [MW_2]^T MW_1$, and denote the singular value decomposition of $S$ as $S = U\Sigma V^T$, for orthogonal matrices $U$ and $V$ and the diagonal matrix $\Sigma$ containing the singular values of $S$ down its diagonal. Similarly, define $\hat{S} = M_2^T M_1$, and denote its singular value decomposition as $\hat{S} = \hat{U}\hat{\Sigma}\hat{V}^T$.

It has been shown in [2] that the $T$ and $\hat{T}$ which minimize (16) and (17) can be expressed as

$$T = UV^T \tag{18}$$
$$\hat{T} = \hat{U}\hat{V}^T. \tag{19}$$

Denote the singular value decomposition of $M$ as $M = L\Psi R^T$, for the matrix with orthornormal columns $L \in \mathbb{R}^{n \times r}$, the orthogonal matrix $R \in \mathbb{R}^{r \times r}$ and the diagonal matrix $\Psi \in \mathbb{R}^{r \times r}$. Writing

$$S = [MW_2]^T MW_1 = W_2^T M^T MW_1 = W_2^T R\Psi L^T L\Psi R^T W_1 = W_2^T R\Psi^2 R^T W_1,$$

makes it clear that $U = W_2^T R$ and $V = W_1^T R$. Therefore

$$T = UV^T = W_2^T RR^T W_1 = W_2^T W_1.$$

To establish the uniqueness of $T$, [2] has shown that it is sufficient that the matrix $S$ have $r$, non-zero singular values, which must be the case since we assume that the rank of $M$ is $r$. Similarly, the uniqueness of $\hat{T}$ follows from the assumption that $\hat{S} = M_2^T M_1$ is of rank $r$.

All that remains is to establish the bound for $\left\| T - \hat{T} \right\|_F$. We first bound $\left\| S - \hat{S} \right\|_F$, and we write

$$\begin{aligned} \left\| S - \hat{S} \right\|_F &= \left\| [MW_2]^T MW_1 - M_2^T M_1 \right\|_F \\ &= \left\| [MW_2]^T MW_1 - [MW_2 + P_2]^T [MW_1 + P_1] \right\|_F. \end{aligned}$$

Note that $\|MW_2\|_F = \|MW_1\|_F = \|M\|_F \leq \delta$. Lemma 12 can now be applied to find

$$\left\| S - \hat{S} \right\|_F \leq \epsilon_1 \delta + \epsilon_2 \epsilon_1 + \epsilon_2 \delta = \delta(\epsilon_1 + \epsilon_2) + \epsilon_1 \epsilon_2. \tag{20}$$

Having established a bound on $\|S - \hat{S}\|$, we now bound $\|T - \hat{T}\|$. Finding $T$ and $\hat{T}$ in (18) and (19) is equivalent to finding the unitary matrix in the polar decomposition of $S$ and $\hat{S}$. Li ([3], theorem 1) has shown if $A$ and $\tilde{A}$ are two non-singular $n \times n$ matrices, then $\left\|\left\| Q - \tilde{Q} \right\|\right\| \leq \frac{2}{\sigma_n(A) + \sigma_n(\tilde{A})} \left\|\left\| A - \tilde{A} \right\|\right\|$, where $Q$ and $\tilde{Q}$ are the unitary matrices in the polar decomposition of $A$ and $\tilde{A}$, $\sigma_n(\cdot)$ gives the $n^{th}$ singular value of it's argument when singular values are indexed such that $\sigma_1 \geq \ldots \geq \sigma_n$ and $\|\|\cdot\|\|$ is any unitarily invariant norm. We can apply this theorem to $S \in \mathbb{R}^{r \times r}$ and $\hat{S} \in \mathbb{R}^{r \times r}$ as we have shown $S = [MW_2]^T MW_1$ is of rank $r$ and by assumption $\hat{S} = M_2^T M_1$ is as well. Applying the theorem for the Frobenius norm, we find

$$\left\| T - \hat{T} \right\|_F \leq \frac{2}{\sigma_r(S) + \sigma_r(\tilde{S})} \left\| S - \tilde{S} \right\|_F \tag{21}$$

$$\leq \frac{2}{\sigma_r(S)} \left\| S - \tilde{S} \right\|_F \tag{22}$$

$$\leq \frac{2}{\sigma_r(S)} [\delta(\epsilon_1 + \epsilon_2) + \epsilon_1 \epsilon_2]. \tag{23}$$

Recognizing that $\sigma_r(S) = \sigma_r([MW_2]^T MW_1) = \sigma_r(M^T M) = \sigma_r^2(M)$, we can complete the proof by writing

$$\left\| T - \hat{T} \right\|_F \leq \frac{2}{\sigma_r^2(M)} [\delta(\epsilon_1 + \epsilon_2) + \epsilon_1 \epsilon_2]. \tag{24}$$

$\square$

**Lemma 17.** *For $C \in \mathbb{R}^{n \times r}$ define $\hat{C}_1$ and $\hat{C}_2$ as*

$$\hat{C}_1 = C(\rho_1, :)W_1 + P_1$$
$$\hat{C}_2 = C(\rho_2 :)W_2 + P_2,$$

*for some index sets $\rho_1 \subseteq [n], \rho_2 \subseteq [n]$ with cardinality $|\rho_1| = n_1$ and $|\rho_2| = n_2$, orthogonal matrices $W_1 \in \mathbb{R}^{r \times r}, W_2 \in \mathbb{R}^{r \times r}$ and error matrices $P_1 \in \mathbb{R}^{n_1 \times r}, P_2 \in \mathbb{R}^{n_2 \times r}$.*

*Denote the intersection of $\rho_1$ and $\rho_2$ as $\iota = \rho_1 \cap \rho_2$. Let $\phi_1$ and $\phi_2$ indicate the rows of $\hat{C}_1$ and $\hat{C}_2$ so that*

$$\hat{C}_1(\phi_1, :) = C(\iota, :)W_1 + P_1$$
$$\hat{C}_2(\phi_2, :) = C(\iota, :)W_2 + P_2.$$

*Finally, define $\eta_2$ such that $\hat{C}_2(\eta_2, :) = C(\rho_2 \setminus \iota, :)W_2 + P_2$.*

---

**Algorithm 1** Procrustes Matrix Reconstruction Algorithm $(\hat{C}_1, \hat{C}_2, \rho_1, \rho_2, \phi_1, \phi_2, \iota, \eta_2)$

---

1. $\hat{T} \leftarrow \operatorname{argmin}_{W:WW^T=I} \left\| \hat{C}_1(\phi_1, :) - \hat{C}_2(\phi_2, :)W \right\|_F$

2. $\hat{C}(\rho_1, :) \leftarrow \hat{C}_1$

3. $\hat{C}(\rho_2 \setminus \iota, :) \leftarrow \hat{C}_2(\eta_2, :)\hat{T}$

4. Return $\hat{C}$

---

*Assume $\rho_1 \cup \rho_2 = [n]$, $|\iota| \geq r$, $\|C(\rho_2, :)\|_F \leq \delta_2$, $\|P_1\|_F \leq \epsilon_1$ and $\|P_2\|_F \leq \epsilon_2$. Then if $C(\iota, :)$, and $\hat{C}_2(\phi_2, :)^T \hat{C}_1(\phi_1, :)$ are of rank $r$, algorithm 1 will assign values to all rows of $\hat{C}$ such that*

$$\min_{W:WW^T=I} \left\| C - \hat{C}W \right\|_F \leq \epsilon_1 + \epsilon_2 + K[\delta_2^2(\epsilon_1 + \epsilon_2) + \delta_2\epsilon_1\epsilon_2],$$

*for a constant $K = \frac{2}{\sigma_r^2(C(\iota, :))}$.*

*Proof.* We first show that all rows of $\hat{C}$ will be assigned values. To see this note that in step 1, rows indexed by $\rho_1$ are assigned values. In step 2, rows indexed by $\rho_2 \setminus \iota$ are assigned values. Finally, $\rho_1 \cup (\rho_2 \setminus \iota) = [n]$, showing that all rows of $\hat{C}$ are assigned values.

To establish the error bound, we begin by using the triangle inequality to write

$$\min_{W:WW^T=I} \left\| C - \hat{C}W \right\|_F \leq \min_{W:WW^T=I} \left\{ \left\| C(\rho_1, :) - \hat{C}(\rho_1, :)W \right\|_F + \left\| C(\rho_2 \setminus \iota, :) - \hat{C}(\rho_2 \setminus \iota, :)W \right\|_F \right\}.$$

If we set $W = W_1^T$, we can write this as

$$\min_{W:WW^T=I} \left\| C - \hat{C}W \right\|_F \leq \left\| C(\rho_1, :) - \hat{C}(\rho_1, :)W_1^T \right\|_F + \left\| C(\rho_2 \setminus \iota, :) - \hat{C}(\rho_2 \setminus \iota, :)W_1^T \right\|_F.$$

We can bound the first term as

$$\begin{aligned}
\left\| C(\rho_1, :) - \hat{C}(\rho_1, :)W_1^T \right\|_F &= \left\| C(\rho_1, :) - \hat{C}_1 W_1^T \right\|_F \\
&= \left\| C(\rho_1, :) - [C(\rho_1, :)W_1 + P_1]W_1^T \right\|_F \\
&= \left\| P_1 W_1^T \right\|_F \\
&= \|P_1\|_F \\
&\leq \epsilon_1,
\end{aligned}$$

where we have used that fact that the Frobenius norm is unitarily invariant in moving from the third to fourth lines. In the remainder of this proof we will use this property of the Frobenius norm without further comment.

We again use the triangle inequality to bound the second term as

$$
\begin{aligned}
\left\| C(\rho_2 \setminus \iota, :) - \hat{C}(\rho_2 \setminus \iota, :) W_1^T \right\|_F &= \left\| C(\rho_2 \setminus \iota, :) - \hat{C}_2(\eta_2, :) \hat{T} W_1^T \right\|_F \\
&= \left\| C(\rho_2 \setminus \iota, :) - [C(\rho_2 \setminus \iota, :) W_2 + P_2(\eta_2, :)] \hat{T} W_1^T \right\|_F \\
&\leq \left\| P_2(\eta_2, :) \hat{T} W_1^T \right\|_F + \left\| C(\rho_2 \setminus \iota, :) \left[ I - W_2 \hat{T} W_1^T \right] \right\|_F
\end{aligned}
$$

We can bound $\left\| P_2(\eta_2, :) \hat{T} W_1^T \right\|_F$ as

$$
\left\| P_2(\eta_2, :) \hat{T} W_1^T \right\|_F = \| P_2(\eta_2, :) \|_F \leq \| P_2 \|_F \leq \epsilon_2,
$$

where we have taken advantage of the fact that $\hat{T} W_1^T$ must be an orthogonal matrix. We now complete the proof by bounding $\left\| C(\rho_2 \setminus \iota, :) \left[ I - W_2 \hat{T} W_1^T \right] \right\|_F$. We can use the submultiplicative property of Frobenius norm to write

$$
\begin{aligned}
\left\| C(\rho_2 \setminus \iota, :) \left[ I - W_2 \hat{T} W_1^T \right] \right\|_F &\leq \| C(\rho_2 \setminus \iota, :) \|_F \left\| I - W_2 \hat{T} W_1^T \right\|_F \\
&= \| C(\rho_2 \setminus \iota, :) \|_F \left\| W_2^T W_1 - \hat{T} \right\|_F \\
&= \| C(\rho_2 \setminus \iota, :) \|_F \left\| T - \hat{T} \right\|_F,
\end{aligned}
$$

where we have defined $T = W_2^T W_1$. We now apply lemma 16 to bound $\left\| T - \hat{T} \right\|_F$. We set $M$ in the lemma to $C(\iota, :)$, $M_1$ to $\hat{C}_1(\phi_1, :)$, $M_2$ to $\hat{C}_2(\phi_2, :)$, $P_1$ to $P_1(\phi_1, :)$ and $P_2$ to $P_2(\phi_2, :)$.

Note that it must be that $\| C(\iota, :) \|_F \leq \| C(\rho_2, :) \|_F$, any by assumption $\| C(\rho_2, :) \|_F \leq \delta_2$ so $\| C(\iota, :) \|_F \leq \delta_2$. Additionally, $\| P_1(\phi_1, :) \|_F \leq \| P_1 \|_F \leq \epsilon_1$ and $\| P_2(\phi_2, :) \|_F \leq \| P_2 \|_F \leq \epsilon_2$. By assumption $C(\iota, :)$ is of rank $r$, as is $\hat{C}_2(\phi_2, :)^T \hat{C}_1(\phi_1, :)$. Under these conditions $T$ is the solution to equation (16) in the lemma and $\hat{T}$ to equation (17). Applying the lemma, we find

$$
\left\| T - \hat{T} \right\|_F \leq \frac{2}{\sigma_r^2(C(\iota, :))} [\delta_2(\epsilon_1 + \epsilon_2) + \epsilon_1 \epsilon_2].
$$

We can also bound $\| C(\rho_2 \setminus \iota, :) \|_F \leq \delta_2$, and we have

$$
\begin{aligned}
\| C(\rho_2 \setminus \iota, :) \|_F \left\| T - \hat{T} \right\|_F &\leq \frac{2}{\sigma_r^2(C(\iota, :))} [\delta_2(\epsilon_1 + \epsilon_2) + \epsilon_1 \epsilon_2] \| C(\rho_2 \setminus \iota, :) \|_F \\
&\leq \frac{2}{\sigma_r^2(C(\iota, :))} [\delta_2(\epsilon_1 + \epsilon_2) + \epsilon_1 \epsilon_2] \delta_2 \\
&= \frac{2}{\sigma_r^2(C(\iota, :))} [\delta_2^2(\epsilon_1 + \epsilon_2) + \delta_2 \epsilon_1 \epsilon_2].
\end{aligned}
$$

Putting all of the pieces of the proof together we have

$$
\min_{W: WW^T = I} \left\| C - \hat{C} W \right\|_F \leq \epsilon_1 + \epsilon_2 + K[\delta_2^2(\epsilon_1 + \epsilon_2) + \delta_2 \epsilon_1 \epsilon_2],
$$

for $K = \frac{2}{\sigma_r^2(C(\iota,:))}$.

$\square$

**Corollary 18.** *For $C \in \mathbb{R}^{n \times r}$, assume there exists a $\hat{C}_1 \in \mathbb{R}^{n_1 \times r}$ and $\hat{C}_2 \in \mathbb{R}^{n_2 \times r}$ and index sets $\rho_1$ and $\rho_2$ such that*

$$\min_{W:WW^T=I} \left\| C(\rho_1,:) - \hat{C}_1 W \right\|_F \leq \epsilon_1$$
$$\min_{W:WW^T=I} \left\| C(\rho_2,:) - \hat{C}_2 W \right\|_F \leq \epsilon_2.$$

Assume $\rho_1 \cup \rho_2 = [n]$. Let $\iota = \rho_1 \cap \rho_2$ and assume $|\iota| \geq r$.

Define $C_l = C(\rho_l,:)$ and let $\phi_l$ be an index set that assigns the rows of the matrix $C(\iota,:)$ to their location in $C_l$, so that $C(\iota,:) = C_l(\phi_l,:)$ and let $\eta_l$ assign the rows of $C(\rho_l \setminus \iota_l,:)$ to their location in $C_l$, so that $C(\rho_l \setminus \iota_l,:) = C_l(\eta_l,:)$.

Assume $C(\iota,:)$ and $\hat{C}_1(\phi_1,:)^T \hat{C}_2(\phi_2,:)$ each are of rank $r$. Finally, assume $\|C(\rho_2,:)\|_F \leq \delta_2$.

Learn $\hat{C} \in \mathbb{R}^{n \times r}$ with algorithm 1 in lemma 17. Then algorithm 1 will assign values to all rows of $\hat{C}$ such that

$$\min_{W:WW^T=I} \left\| C - \hat{C}W \right\|_F \leq \epsilon_1 + \epsilon_2 + K[\delta_2^2(\epsilon_1 + \epsilon_2) + \delta_2 \epsilon_1 \epsilon_2],$$

for $K = \frac{2}{\sigma_r^2(C(\iota,:))}$.

*Proof.* For the $r \times r$ orthogonal matrices $T_1$ and $T_2$, define the matrices $E_1, E_2 \in \mathbb{R}^{n \times r}$ such that

$$\hat{C}_1 = C(\rho_1,:)T_1 + E_1 \tag{25}$$
$$\hat{C}_2 = C(\rho_2,:)T_2 + E_2. \tag{26}$$

Then for any $\hat{C}_1, \hat{C}_2$ and $C$, there exists orthogonal $T_1$ and $T_2$ such that $\|E_1\|_F \leq \epsilon_1$ and $\|E_2\|_F \leq \epsilon_2$. We prove this first for $\hat{C}_1$ and $C(\rho_1,:)$. Let $W' = \operatorname{argmin}_{W:WW^T=I} \left\| C(\rho_1,:) - \hat{C}_1 W \right\|_F$, and let $T_1 = W'^T$. Then

$$
\begin{aligned}
\|E_1\|_F &= \left\| \hat{C}_1 - C(\rho_1,:)W'^T \right\|_F \\
&= \left\| C(\rho_1,:) - \hat{C}_1 W' \right\|_F \\
&= \min_{W:WW^T=I} \left\| C(\rho_1,:) - \hat{C}_1 W \right\|_F \\
&\leq \epsilon_1.
\end{aligned}
$$

The same argument can be carried out for $\hat{C}_2$, $C(\rho_2,:)$ and $T_2$. All of the assumptions are then met for lemma 17 to guarantee that algorithm 1 will learn a $\hat{C}$ from $\hat{C}_1$ and $\hat{C}_2$ such that all rows of $\hat{C}$ are assigned and

$$\min_{W:WW^T=I} \left\| C - \hat{C}W \right\|_F \leq \epsilon_1 + \epsilon_2 + K[\delta_2^2(\epsilon_1 + \epsilon_2) + \delta_2 \epsilon_1 \epsilon_2],$$

for $K = \frac{2}{\sigma_r^2(C(\iota,:))}$.

$\square$

**Corollary 19.** *For $C \in \mathbb{R}^{n \times r}$, assume there exists a set of two or more matrices $\hat{C}_1 \in \mathbb{R}^{n_1 \times r}, \ldots, \hat{C}_k \in \mathbb{R}^{n_k \times r}$, and collection of index sets $\rho_1, \ldots, \rho_k$ such that $\|C(\rho_l, :)\|_F \leq b$ for some $b > 0$ and $\min_{W:WW^T=I} \left\|C(\rho_l, :) - \hat{C}_l W\right\|_F \leq \epsilon/b$ some $\epsilon \geq 0$ for all $l$.*

*Assume $\cup_{l=1}^k \rho_l = [n]$. For all $l \geq 2$, define $\iota_l = \rho_l \cap \left(\cup_{j=1}^{l-1} \rho_j\right)$ and assume $|\iota_l| \geq r$.*

*Define $C_l = C(\rho_l, :)$ for all $l$, and let $\phi_l$ be an index set that assigns the rows of the matrix $C(\iota, :)$ to their location in $C_l$, so that $C(\iota, :) = C_l(\phi_l, :)$ and let $\eta_l$ assign the rows of $C(\rho_l \setminus \iota_l, :)$ to their location in $C_l$, so that $C(\rho_l \setminus \iota_l, :) = C_l(\eta_l, :)$.*

---

**Algorithm 2** Sequential Procrustes Matrix Recovery $(\hat{C}_1, \ldots, \hat{C}_k, \rho_1, \ldots, \rho_k, \{\iota_l, \phi_l, \eta_l\}_{l=2}^k)$

---

Initialize $\hat{C}$ as a $n \times r$ matrix.

1. $\hat{C}(\rho_1, :) \leftarrow \hat{C}_1$

2. For $l \in \{2, \ldots, k\}$

    (a) $\hat{W}_l \leftarrow \operatorname{argmin}_{WW^T=I} \left\|\hat{C}(\iota_l, :) - \hat{C}_l(\phi_l, :)W\right\|_F^2$

    (b) $\hat{C}(\rho_l \setminus \iota_l, :) \leftarrow \hat{C}_l(\eta_l, :)\hat{W}_l$

3. Return $\hat{C}$

---

*Use algorithm 2 to learn a $\hat{C} \in \mathbb{R}^{n \times r}$. Then if $C(\iota_l, :)$ and $\hat{C}_l(\phi_l, :)^T \hat{C}(\iota_l, :)$ each are of rank $r$ for $l \in 2, \ldots, k$, algorithm 2 will assign values to all the rows of $\hat{C}$ such that*

$$\min_{W:WW^T=I} \left\|C - \hat{C}W\right\| \leq [4 + 12/v]^{k-1}\epsilon,$$

*for a constant where $v \leq \sigma_r^2(C(\iota_l, :))/b^2$ for all $l \geq 2$.*

*Proof.* We will first establish a bound assuming $\|C(\rho_l, :)\|_F \leq 1$ for all $l$ and $\epsilon \leq 1$, and will then generalize this result when $\|C(\rho_l, :)\|_F \leq b$ for some $b > 0$ for all $l$ and $\epsilon \leq b$.

We establish the result by induction. We prove the base case for $k = 2$. Note for the case $k = 2$, algorithm 2 will consist of step 1, followed by one iteration of step 2, followed by step 3, which is equivalent to running algorithm 1 of lemma 17 with inputs $\hat{C}_1, \hat{C}_2, \rho_1, \rho_2, \phi_1, \phi_2, \iota_2$ and $\eta_2$, where we have listed inputs in the same order as in the heading for algorithm 1. In this case, $\hat{C}_1$ and $\hat{C}_2$ are such that $\min_{W:WW^T=I} \left\|C(\rho_1, :) - \hat{C}_1 W\right\| \leq \epsilon$ and $\min_{W:WW^T=I} \left\|C(\rho_2, :) - \hat{C}_2 W\right\| \leq \epsilon$. Further, by assumption $\rho_1 \cup \rho_2 = [n]$, $|\iota_2| \geq r$ and $C(\iota_2, :)$ is of rank $r$. If $\hat{C}_2(\phi_2, :)^T \hat{C}_1(\phi_1, :)$ is of rank $r$, corollary 18 then guarantees that algorithm 1 will learn a $\hat{C}$ such that

$$\min_{W:WW^T=I} \left\|C - \hat{C}W\right\|_F \leq 2\epsilon + K[2\delta_2^2\epsilon + \delta_2\epsilon^2], \tag{27}$$

for $K = \frac{2}{\sigma_r^2(C(\iota_2, :))}$ where $\delta_2 = \|C(\rho_2, :)\|_F \leq 1$. Noting that by assumption $\epsilon \leq 1$ and $\sigma_r^2(C(\iota_2, :)) \leq v$, we have

$$\min_{W:WW^T=I} \left\|C - \hat{C}W\right\|_F \leq [2 + 3G]\epsilon \leq [4 + 6G]\epsilon, \tag{28}$$

where $G = 2/v$.

We now prove the induction step. When $k \geq 3$, algorithm 2 will consist of running step 1, followed by $k - 1$ iterations of step 2, followed by running step 3. Let $\mathcal{I}_{k-1} = \cup_{l=1}^{k-1} \rho_l$. On the $(k-1)^{th}$ iteration of step 2, assume that the previous steps of the algorithm have learned the rows of $\hat{C}$ indexed by $\mathcal{I}_{k-1}$ (that is $C(\mathcal{I}_{k-1}, :)$) such that $\min_{W:WW^T=I} \left\| C(\mathcal{I}_{k-1}, :) - \hat{C}(\mathcal{I}_{k-1}, :)W \right\| \leq [4 + 6G]^{k-2}\epsilon$. For clarity of what is to follow, define $\epsilon'_{k-1} = [4 + 6G]^{k-2}\epsilon$. Performing the $(k-1)^{th}$ iteration of step 2 of algorithm 2 with $\hat{C}_k$ and the $\hat{C}(\mathcal{I}_{k-1}, :)$ from the previous steps of the algorithm and then performing step 3 is equivalent to running algorithm 1 with the inputs $\hat{C}(\mathcal{I}_{k-1}, :), \hat{C}_k, \mathcal{I}_{k-1}, \rho_k, \phi'_{k-1}, \phi_k, \eta_k, \iota_k$, where we have again listed inputs in the same order as in the heading for algorithm 2, and $\phi'_{k-1} = \iota_k$.

By assumption $\min_{W:WW^T=I} \left\| C(\mathcal{I}_{k-1}, :) - \hat{C}'_{k-1}W \right\|_F \leq \epsilon'_{k-1}$ and $\min_{W:WW^T=I} \left\| C(\rho_k, :) - \hat{C}_k W \right\|_F \leq \epsilon$. Further, $\rho_k \cup \mathcal{I}_{k-1} = \rho_k \cup (\cup_{l=1}^{k-1} \rho_l) = [n]$, $|\iota_k| \geq r$ and $C(\iota_k, :)$ is of rank $r$. Then if $\hat{C}_k(\phi_k, :)^T \hat{C}(\iota, :)$ also is of rank $r$, corollary 18 guarantees that algorithm 1 will learn a full $\hat{C}$ from $\hat{C}_k$ and $\hat{C}'_{k-1}$ such that

$$\min_{W:WW^T=I} \left\| C - \hat{C}W \right\|_F \leq \epsilon + \epsilon'_{k-1} + K[\delta_k^2(\epsilon + \epsilon'_{k-1}) + \delta_k \epsilon'_{k-1}\epsilon],$$

where $\delta_k = \|C(\rho_k, :)\|_F \leq 1$. Again, $\epsilon \leq 1$, so we have

$$\min_{W:WW^T=I} \left\| C - \hat{C}W \right\|_F \leq \epsilon + \epsilon'_{k-1} + K[\epsilon + 2\epsilon'_{k-1}].$$

Recognizing that again $K = \frac{2}{\sigma_r^2(C(\iota_k, :))} \leq \frac{2}{v} = G$, we further have

$$\min_{W:WW^T=I} \left\| C - \hat{C}W \right\|_F \leq \epsilon + \epsilon'_{k-1} + G[\epsilon + 2\epsilon'_{k-1}]. \tag{29}$$

We can bound the right hand side of (29) as

$$
\begin{aligned}
\min_{W:WW^T=I} \left\| C - \hat{C}W \right\|_F &\leq \epsilon + [4 + 6G]^{k-2}\epsilon + G[\epsilon + 2[4 + 6D]^{k-2}\epsilon] \\
&= [1 + G]\epsilon + [1 + 2G][4 + 6G]^{k-2}\epsilon \\
&= [1 + G]\epsilon + 2^{k-2}[1 + 2G][2 + 3G]^{k-2}\epsilon \\
&\leq [2 + 3G]\epsilon + 2^{k-2}[2 + 3G]^{k-1}\epsilon \\
&= \left([2 + 3G] + 2^{k-2}[2 + 3G]^{k-1}\right)\epsilon \\
&\leq \left([2 + 3G]^{k-1} + 2^{k-2}[2 + 3G]^{k-1}\right)\epsilon \\
&= [1 + 2^{k-2}][2 + 3G]^{k-1}\epsilon \\
&\leq [2^{k-2} + 2^{k-2}][2 + 3G]^{k-1}\epsilon \\
&= 2^{k-1}[2 + 3G]^{k-1}\epsilon \\
&= [4 + 6G]^{k-1}\epsilon. \tag{30}
\end{aligned}
$$

Substituting $G = 2/v$, we can conclude

$$\min_{W:WW^T=I} \left\| C - \hat{C}W \right\|_F \leq [4 + 12/v]^{k-1}\epsilon$$

when $\|C(\rho_l, :)\|_F \leq 1$ for all $l$ and $\epsilon \leq 1$.

We now generalize this result when $\|C(\rho_l,:)\|_F \leq b$ for some $b > 0$ for all $l$ and $0 \leq \epsilon \leq b$. Define $C' = C/b$ and $\hat{C}'_l = \hat{C}_l/b$ for all $l$. Then

$$\|C'(\rho_l,:)\|_F = \|C(\rho_l,:)/b\|_F = \frac{1}{b}\|C(\rho_l,:)\|_F \leq 1.$$

Further, since $\min_{W:WW^T=I} \left\|C(\rho_l,:) - \hat{C}_l W\right\|_F \leq \epsilon$ and $\epsilon \leq b$, then

$$
\begin{aligned}
\min_{W:WW^T=I} \left\|C'(\rho_l,:) - \hat{C}'_l W\right\|_F &= \min_{W:WW^T=I} \left\|\frac{1}{b}C(\rho_l,:) - \frac{1}{b}\hat{C}_l W\right\|_F \\
&= \frac{1}{b}\left[\min_{W:WW^T=I} \left\|C(\rho_l,:) - \hat{C}_l W\right\|_F\right] \\
&\leq \frac{\epsilon}{b} \\
&\leq 1.
\end{aligned}
$$

for all $l$. For clarity of what is to follow define $\epsilon' = \epsilon/b$.

Finally, if $v \leq \sigma_r^2(C(\iota_l,:))$ for all $l \geq 2$, then $v' = v/b^2$ will satisfy $v' \leq \sigma_r^2(C'(\iota_l,:))$ for all $l \geq 2$, as can be seen by writing

$$\frac{v}{b^2} \leq \frac{1}{b^2}\sigma_r^2(C(\iota_l,:)) = \sigma_r^2(C(\iota_l,:)/b) = \sigma_r^2(C'(\iota_l,:)).$$

Having established this basic inequalities, it can be verified that using algorithm 2 to reconstruct $C$ from $\hat{C}_1,\ldots,\hat{C}_k$ is equivalent to first using algorithm 2 to reconstruct the rescaled $C'$ from $\hat{C}'_1,\ldots,\hat{C}'_k$ and then multiplying the final $\hat{C}'$ by $b$ to estimate $\hat{C} = b\hat{C}'$. However, we have just shown that $\|C'(\rho_l,:)\|_F \leq 1$ for all $l$ and $\epsilon' \leq 1$, so we can use the result we just established to conclude that

$$\min_{W:WW^T=I} \left\|C' - \hat{C}'W\right\|_F \leq [4 + 12/v']^{k-1}\epsilon'$$

for $v' \leq \sigma_r^2(C'(\iota_l,:))$ for all $l$. Using the relations above, we can conclude

$$\min_{W:WW^T=I} \left\|C' - \hat{C}'W\right\|_F \leq [4 + 12/v']^{k-1}\frac{\epsilon}{b}$$

when for $v' \leq \frac{1}{b^2}\sigma_r^2(C(\iota_l,:))$ for all $l$.

However, if $\min_{W:WW^T=I} \left\|C' - \hat{C}'W\right\|_F \leq [4 + 12/v']^{k-1}\frac{\epsilon}{b}$, then

$$
\begin{aligned}
b\left[\min_{W:WW^T=I} \left\|C' - \hat{C}'W\right\|_F\right] &= \min_{W:WW^T=I} \left\|bC' - b\hat{C}'W\right\|_F \\
&= \min_{W:WW^T=I} \left\|C - \hat{C}W\right\|_F \\
&\leq b[4 + 12/v']^{k-1}\frac{\epsilon}{b} \\
&= [4 + 12/v']^{k-1}\epsilon,
\end{aligned}
$$

so $\min_{W:WW^T=I} \left\|C - \hat{C}W\right\|_F \leq [4 + 12/v']^{k-1}\epsilon$ which completes the proof.

$\square$

## Footnotes

[1]By this indexing notation, we count iterations of step 2 starting with the number 2. For example, if step two runs twice, then for the first iteration of step 2, $l = 2$, and for the second $l = 3$.

[2] We will use the notation $\{\cdot\} + \{\cdot\}$ indicates set addition.

[3]Algorithm 1 never explicitly forms $\hat{C}_1$, but for clarify and brevity, we will define $\hat{C}(\rho_{\tau_l}, :)$ to be $\hat{C}(\rho_{\tau_l}, :) = \hat{C}_1$.