[Reviews · NeurIPS 2014]

Submitted by Assigned_Reviewer_35

[Summary]
Reconstruction of a symmetric positive semi-definite matrix is studied
under the assumption that the matrix elements are sampled on a given
collection of the principal minors. The necessary and sufficient
conditions for the exact reconstruction in the noiseless case have
been provided (Theorems 1 and 2), an algorithm to solve a problem
instance has been proposed, and a non-probabilistic bound on estimation
error in the noisy case has been proved (Theorem 4).

[Quality]
The problem seemed at first not so interesting, but the author has studied
it thoroughly, and has eventually arrived at a nontrivial non-probabilistic
bound on estimation error (Theorem 4), which is very interesting.

[Clarity]
This paper is very clearly written and quite easy to read.

[Originality]
The problem studied in this paper should be original,
since as far as my knowledge it has not been studied before.
The problem is also motivated by real-world problems, which
makes this paper more appealing.

[Significance]
The study has been motivated by real-world applications. Necessary
and sufficient conditions have been obtained. The proposed algorithm
has been proved to solve all problems which should be solvable in the
noiseless case. A non-probabilistic bound on error in the estimate
has been obtained in the noisy case. These aspects make the paper
very significant.
Summary: Reconstruction of a symmetric positive semidefinite matrix has been studied under a unique, but well-motivated, setting, to obtain non-trivial and plausible results. This paper thus reports an interesting piece of work.

Submitted by Assigned_Reviewer_37

This paper considers the problems of completing a positive semidefinite matrix when the observations are in the forms of principal minors. The proposed algorithm is quite natural and intuitive, and is shown to succeed under minimal deterministic conditions. A stability analysis is also given when the observations are noisy. The paper is well written and the presentation is clear. My main concerns are there might be some limitations with the results in the paper:

(1) It is not immediately clear that given the full set Omega of observed entries, how to extract and order the principal minors Omega_k such that the assumption (A3) is satisfied. Is this problem trivial, or does it requires some intelligence? This issue is not addressed explicitly in Algorithm 1 and the experiments.

(2) The algorithm and its proof seem to rely heavily on the assumption that the observations are exactly in the form of principal minors (or a superset of it), and the minors have sufficient overlaps. It is unclear if the performance is sensitive to deviations in the observation pattern.

(3) The requirement on the observation pattern means either the observations have to come naturally in the form of principal minors with overlaps, or the practitioner must have complete control over the sampling process and can query the entries in this particular way. This might limit the applicability of the algorithm. It would be more convincing if the author can present experiments on at least one real-world dataset which meets such sampling requirement.

(4) The analysis in Theorem 4 does not provide much information beyond that the error goes to zero as the noise goes to zero (which is not too surprising as the problem is over-determined). The error bound depends exponentially on k and involves powers of the eigen gaps, so the bound might be quite large. Experiment is given for noisy completion but for a limited setting with matrices of tiny sizes.
Summary: The paper considers an interesting problem, and the proposed solution is intuitive and comes with guarantees. However, the results might have several issues that need to be addressed, and the experiments only consider very small problems.

Submitted by Assigned_Reviewer_45

PAPER SUMMARY

In this paper, the authors propose a model and an algorithm for completing a symmetric, positive semidefinite matrix. Unlike prior work that assumes that the observed entries are drawn uniformly at random, this work assumes that the observed entries come from deterministic collection of its principal sub matrices. The paper derives either necessary or sufficient conditions on these sub matrices that guarantee the perfect recovery of the unknown entries. The paper also considers the case where the data is corrupted by noise and derives an upper bound on the error between the estimated and the true matrix. Simulations on synthetic data verify the correctness of the proposed model.

SPECIFIC COMMENTS

This paper is well-organized and mathematically rigorous. The theoretical analysis is novel since it addresses deterministic matrix completion model with specific structure. In particular, rather than addressing the well studied case where the observed entries are drawn uniformly at random, the paper assumes that the observed entries are drawn from a set of principal sub matrices that satisfy certain properties. Perfect matrix completion can be achieved in the noiseless case, while an upper bound is derived for the noisy case.

Overall, I find the results in this paper to be very novel. I only have a few minor comments below that can help improve clarity. My most important comment is that the experiments can be improved by adding a comparison with other methods as well as experiments on real data.

1) The term "minor" usually refers to the "determinant of some smaller square matrix". In this paper, the term is used to refer to the sub matrix itself without the determinant. I suggest the authors use a different name to avoid confusion.

2) Some of the notation is confusing. For example, C_l and C(\rho_l, :) are both used to refer to the same matrix, which is confusing to the reader.

3) The proposed model and algorithm are based on several assumptions on the principal sub matrices of the data matrix. In practice, it is not clear if these assumptions are satisfied in real world applications. For example, given a dataset, how does one derive the collection of principal sub matrices that satisfy assumptions A1-A3? Or if these assumptions fail, is there a way of permuting the rows and columns of the given matrix so that the assumptions are satisfied?

4) In the experiments, the authors did simulations on synthetic data to verify the correctness of the algorithm. However there is no comparison with other matrix completion methods, which is not convincing enough to show the advantage of the proposed algorithm. It would have been nice to see an example where standard matrix completion algorithms fail, but the propose algorithm succeeds. Also, the paper would be strengthened if there were simulations on real data. One specific example that comes to mind is motion segmentation in computer vision, where the missing entries occur due to appearance and disappearance of moving objects in a video. In that case, principal sub matrix patterns show up naturally.
Summary: This paper is well-organized and mathematically rigorous. The theoretical analysis is novel since it addresses deterministic matrix completion model with specific structure. The experiments can be improved by adding a comparison with other methods as well as experiments on real data.
Author Feedback
Author rebuttal: We thank the reviewers for their time and helpful comments and suggestions.

1. Reviewers 37 and 45 ask if assumptions A1-A3 are met in practice on real-world datasets. There are indeed real-world scenarios where these assumptions are met. For example, in neuroscience overlapping subpopulations of neurons that researchers sequentially record from naturally define overlapping principal submatrices of the covariance matrix for the whole population. While researchers do not currently have the fidelity of control to determine the individual makeup of a population of recorded neurons with present day technology (e.g., calcium imaging), recording devices can be moved in a gross manner so as to ensure some overlap of recorded populations. While we are limited by space in the present paper to fully explore this idea, we agree the final paper would be strengthened by providing a discussion of at least one specific, real-world example where we show how exactly a set of principal submatrices meeting assumptions A1 - A3 would be derived based on the way data is collected in a real-world scenario. We will include such a discussion in the final manuscript.

2. Reviewer 37 asks how to extract and order a set of principal submatrices from the set, Omega, of observed entries. This is a very interesting question for future research. To the best of our knowledge the computational complexity of the fully general problem (finding and ordering a set of principal submatrices that meet assumptions A1-A3 from an arbitrary set of revealed entries with no information about how entries were revealed) is unknown. However, in many scenarios, such as the neuroscience example above, available information about the way data was collected facilitates the identification of the set of principal submatrices. In the final version of the paper, we will add a discussion of these points.

3. Reviewer 45 also notes that the paper would be strengthened by comparing the performance of our method to that of other methods. We fully agree. While we feel doing this justice will require a full length journal paper, we expect the algorithm in the present paper to have at least two distinct benefits over previous work. First, as we require no incoherence assumptions, it should be possible to construct a matrix and a set of revealed entries for which our method succeeds but most existing methods fail. Second, many existing matrix completion algorithms, such as those which solve a nuclear norm minimization problem, can be computationally expensive. By performing singular value decomposition on a set of small principal submatrices and then stitching the results together, our algorithm has the potential to be more computationally efficient than existing methods when the individually revealed principal submatrices are small compared to the size of the full matrix.

4. We thank reviewer 45 for pointing out the standard usage of principal minor and places where we can clarify our notation. We will incorporate these comments into the final version of the manuscript.